

# Entanglement dynamics with string measurement operators

**Giulia Piccitto**[1,2]⋆, **Angelo Russomanno**[3,4] **and Davide Rossini**[1]

**1** Dipartimento di Fisica dell'Università di Pisa and INFN,
Largo Pontecorvo 3, I-56127 Pisa, Italy
**2** Dipartimento di Matematica e Informatica, Università di Catania,
Viale Andrea Doria 6, 95125, Catania, Italy
**3** Scuola Superiore Meridionale, Università di Napoli Federico II,
Largo San Marcellino 10, I-80138 Napoli, Italy
**4** Dipartimento di Fisica "E. Pancini," Università di Napoli
Federico II, Monte S. Angelo, I-80126 Napoli, Italy

⋆ giulia.piccitto@unict.it

## Abstract

We explain how to apply a Gaussian-preserving operator to a fermionic Gaussian state. We use this method to study the evolution of the entanglement entropy of an Ising spin chain undergoing a quantum-jump dynamics with string measurement operators. We find that, for finite-range string operators, the asymptotic entanglement entropy exhibits a crossover from a logarithmic to an area-law scaling with the system size, depending on the Hamiltonian parameters as well as on the measurement strength. For ranges of the string which scale extensively with the system size, the asymptotic entanglement entropy shows a volume-law scaling, independently of the measurement strength and the Hamiltonian dynamics. The same behavior is observed for the measurement-only dynamics, suggesting that measurements may play a leading role in this context.

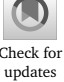

# 1  Introduction

Understanding the role of entanglement in quantum many-body systems is a subject under intensive study [1,2]. While the ground-state entanglement properties have been widely investigated and are relatively well understood, there are still several open questions regarding the stabilization of quantum correlations during the non-equilibrium dynamics. By exploiting a quasi-particle description it has been shown that in integrable systems, after a sudden quench, the entanglement entropy linearly increases in time, and eventually settles to a stationary value which linearly scales with the system size (volume-law scaling) [3,4]. A similar volume-law scaling occurs in ergodic short-range systems [5], while in the presence of a strong disorder the situation is more involved, due to the appearance of many-body localization (see Ref. [6] for a review).

A natural extension of the above scenario is for a quantum system coupled to an external environment, a setup which is crucial to applications in quantum technologies and quantum computing, since any experimental platform is subject to noise-induced decoherence [7, 8]. In the hypothesis of weak and Markovian coupling with the bath, a good modeling for this setup is to assume the evolution of the reduced density matrix of the system to be ruled by a master equation in a Lindblad form [9]. The system-bath framework can be also interpreted as a process in which the environment behaves as a classical stochastic process performing random measurements on the quantum system [10, 11]. The outcome of any realization of this process is a pure-state stochastic quantum trajectory and the density matrix obtained by averaging over such trajectories obeys a Lindblad-type dynamic evolution [12, 13].

Here we are interested in the so-called measurement-induced phase transitions (MIPTs), i.e., in the possibility to develop discontinuities as in the asymptotic entanglement properties that emerge from the interplay between the unitary dynamics (typically generating entanglement) and the measurement processes (typically destroying it). This kind of transitions, as well as other peculiar singularities emerging from quantum measurements, have been first proposed and studied in unitary random circuits undergoing random measurements (see, e.g., Refs. [14–29]), which may display a transition between a phase in which the entanglement grows with the system size and another phase where it follows an area-law scaling (i.e., it saturates to a finite and constant value, for one dimensional systems). Similarly, MIPTs may also arise in free-fermion random circuits with temporal randomness [30], Majorana random circuits [31, 32], Dirac fermions [33], non-Hermitian topological systems [34–37] and other Hamiltonian models undergoing random measurements at discrete times [38–41]. When mea-

suring string operators, MIPTs can occur in the presence of a measurement-only dynamics, without the interplay with the Hamiltonian [42, 43].

Recently, an increasing number of works have been focusing on MIPTs in fermionic systems described by quadratic Hamiltonians on a lattice, undergoing quantum trajectories under the action of random measurements of onsite quadratic operators [44–61]. One of the reasons of this interest relies in the Gaussian structure of the states of such systems that, being entirely determined by two-point correlation functions, are suitable for a semi-analytical treatment up to large lattice sizes (of the order of hundreds of sites) [62–68]. In particular, it has been found that the asymptotic volume-law scaling of the entanglement following a unitary evolution [69, 70] is unstable under local measurements, independently of the measurement strength [44, 46], and a MIPT may emerge. For example, the Ising chain exhibits a size-dependent crossover towards a subextensive or an area-law entanglement regime, depending on the system parameters, as well as on the measurement rate and the type of unraveling [47–51, 71]. In a version of this model with temporal noise also in the Hamiltonian, this crossover has been analytically proved to be a true transition by means of the replica method [72, 73]. Similarly, it has been found that entanglement transitions can arise, even in the absence of a coherent evolution, from the competition between different measurement protocols [74]. In the same context of monitored fermionic models, a couple of recent works pointed out the importance of having long-range interactions to stabilize larger amounts of entanglement, if the system undergoes random onsite measurements [53, 54] (see [75, 76] for a similar phenomenon in quantum circuits). Similar results have been found also for short-ranged Hamiltonian in the presence of long-range continuous measurement processes [77].

Inspired by these latter findings, here we follow a complementary strategy and address the role of non-local string measurement operators in the dynamics of entanglement for a quadratic Hamiltonian system. The purpose of this paper is twofold. On the one hand, we collect and review all the technical details to implement the dynamics of Gaussian states in the presence of Gaussian-preserving measuring operators. On the other hand, we discuss the effect of non-local measurement operators on the dynamics of the entanglement entropy, while keeping the Hamiltonian with short-range interactions. To this aim, we implement a quantum-jump unraveling of the Lindblad dynamics [9, 10] for the Ising chain (mapped to the fermionic Kitaev chain) using strings of Pauli matrices as Lindblad operators (mapped to non-local two-point fermionic operators). The main result is that, when the string range $r$ is comparable with the system size $L$, contrary to what happens in the presence of local jump operators, the measurement process can stabilize an asymptotic volume-law for the entanglement entropy. As a consequence, we cannot exclude that a MIPT can occur as a function of $r$. Quite interestingly, the same qualitative conclusion applies if we switch off the Hamiltonian and consider a measurement-only Lindblad dynamics induced by string operators, similarly to what has been studied in Ref. [42] for measurement-only models with nonlocal measurements of Pauli strings.

The paper is organized as follows. In Section 2 we briefly review the theory of quadratic fermionic Hamiltonians on a lattice. In Section 3 and Appendix A we present the technical details required to implement the dynamics of a Gaussian state under the effect of Gaussian-preserving exponentials of quadratic operators. We first set the general framework, and then shed light on the dynamical equations and on the normalization of the state. In Section 4 we apply such a machinery to string operators, a particular class of operators which generalize the one discussed in Ref. [49, 50]. Subsequently, in Section 5 we present and discuss numerical results for the Ising chain with non-local string measurement operators. Finally, our conclusions are drawn in Section 6. The other appendices contain some details on the quantum-jump measurement protocol (Appendix B) and an alternative approach to determine the evolved state with the string operator (Appendix C).

## 2   Fermionic Gaussian states

We consider a generic free-fermion system on a lattice with $L$ sites, described by the quadratic Hamiltonian

$$\hat{H} = \sum_{i,j=1}^{L} \left( Q_{i,j}\, \hat{c}_i^\dagger \hat{c}_j + P_{i,j}\, \hat{c}_i^\dagger \hat{c}_j^\dagger + \text{h.c.} \right), \tag{1}$$

where $\hat{c}_j^{(\dagger)}$ denotes the anticommuting annihilation (creation) operator on the $j$-th site. To ensure the Hermiticity of $\hat{H}$, the coefficient matrices obey $Q^\dagger = Q$ and $P^T = -P$.

In this context, a Gaussian state has the generic form[1]

$$|\psi\rangle = \mathcal{N} \exp\left\{ \frac{1}{2} \sum_{p,q} Z_{p,q}\, \hat{c}_p^\dagger \hat{c}_q^\dagger \right\} |0\rangle_{\hat{c}}, \tag{2}$$

with

$$Z = -\left(u^\dagger\right)^{-1} v^\dagger, \tag{3}$$

and $\mathcal{N}$ a normalization constant. Here $u$ and $v$ are $L \times L$ matrices that obey the following symplectic unitarity conditions

$$u^\dagger u + v^\dagger v = I, \qquad u v^\dagger + v^* u^T = 0, \tag{4}$$

where $I$ denotes the $L \times L$ identity matrix. These conditions are equivalent to say that the matrix

$$\mathbb{U} = \begin{pmatrix} u & v^* \\ v & u^* \end{pmatrix}, \tag{5}$$

is unitary and preserves the fermionic anticommutation relations: In fact it defines a new set of fermionic anticommuting operators

$$\hat{\gamma}_\kappa = \sum_j \left( u_{j,\kappa}^* \, \hat{c}_j + v_{j,\kappa}^* \, \hat{c}_j^\dagger \right), \tag{6}$$

that enjoy the property

$$\hat{\gamma}_\kappa |\psi\rangle = 0, \qquad \forall\, \kappa \in \{1, \dots, L\}, \tag{7}$$

meaning that $|\psi\rangle$ is the vacuum of the $\hat{\gamma}$ fermions, $|\psi\rangle \equiv |0\rangle_{\hat{\gamma}}$ (the matrix $\mathbb{U}$ is the so-called Bogoliubov transformation. For more details see, e.g., Refs. [66,78]).

We remark that, because of the fermionic nature of the $\hat{c}$-operators, the matrix $Z$ in Eq (3) is antisymmetric, as can be proved by exploiting Eqs. (4) [78]. In this context, the Nambu spinor notation is useful. Defining $\hat{\Psi} \equiv (\hat{c}_1, \cdots, \hat{c}_L, \hat{c}_1^\dagger, \cdots, \hat{c}_L^\dagger)^T$, and $\hat{\Phi} \equiv (\hat{\gamma}_1, \cdots, \hat{\gamma}_L, \hat{\gamma}_1^\dagger, \cdots, \hat{\gamma}_L^\dagger)^T$, one can rewrite Eq. (6) as

$$\hat{\Phi} = \mathbb{U}^\dagger \hat{\Psi}. \tag{8}$$

The state in Eq. (2) is Gaussian, enjoys the Wick's theorem, and thus it is completely determined by two-point correlation functions. Using Eq. (7) one can show that these correlation functions have a simple expression in terms of the $\mathbb{U}$ matrix:

$$G_{i,j} \equiv \langle \hat{c}_i^\dagger \hat{c}_j \rangle = \left[ v\, v^\dagger \right]_{i,j}, \qquad F_{i,j} \equiv \langle \hat{c}_i \hat{c}_j \rangle = \left[ v\, u^\dagger \right]_{i,j}. \tag{9}$$

An important case where the state has the Gaussian form in Eq. (2) is the ground state of the Hamiltonian (1). The Hamiltonian reads

$$\hat{H} = \frac{1}{2} \hat{\Psi}^\dagger \mathbb{H} \hat{\Psi} + \text{const.}, \quad \text{with} \quad \mathbb{H} = \begin{pmatrix} Q & P \\ -P^* & -Q^* \end{pmatrix}. \tag{10}$$

---

[1]Hereafter we omit extrema in all the summations running from 1 to $L$.

The $2L \times 2L$ Hermitian matrix $\mathbb{H}$ is called the Bogoliubov-de Gennes Hamiltonian matrix, which can be diagonalized through the implementation of a Bogoliubov transformation $\mathbb{U}$ of the form (5), such that $\mathbb{U}^{-1}\mathbb{H}\mathbb{U} = \text{diag}[\omega_\kappa, -\omega_\kappa]_{\kappa=1,\cdots,L}$. The above transformation defines a new set of fermionic quasi-particles $\hat{\gamma}_\kappa$ [Eq. (6)], according to which the Hamiltonian of Eq. (1) can be written as

$$\hat{H} = \sum_\kappa \omega_\kappa \hat{\gamma}_\kappa^\dagger \hat{\gamma}_\kappa + \text{const.} \tag{11}$$

Its ground state is a Gaussian state of the form in Eq. (2), that corresponds to the vacuum of the $\hat{\gamma}_\kappa$ operators [see Eq. (7)].

We note that the above formalism can be easily generalized to the case of time-varying coefficients $Q_{i,j}(t)$ and $P_{i,j}(t)$ in Eq. (1), by admitting a time dependence of the Bogoliubov transformation matrix $\mathbb{U} = \mathbb{U}(t)$. Moreover, as we shall see below in Section 3, the form of a generic Gaussian state Eq. (2) is preserved by the application of the exponential of any operator quadratic in the $\hat{c}$ fermions.

## 3 Evolving Gaussian states

We now discuss the evolution of a generic Gaussian state $|\psi\rangle$ under the effect of an operator of the form

$$\hat{M}(s) = e^{\xi s \hat{A}}, \tag{12a}$$

being $s \in \mathbb{R}$, $\xi = \{1, i\}$, and

$$\hat{A} = \sum_{i,j} \left( D_{i,j}\, \hat{c}_i^\dagger \hat{c}_j + O_{i,j}\, \hat{c}_i^\dagger \hat{c}_j^\dagger + \text{h.c.} \right), \tag{12b}$$

a generic Hermitian quadratic operator. For simplicity of notations, we assume all coefficients in $\hat{A}$ to be real, but our analysis can be easily generalized to the complex case.

Due to the fermionic anticommutation rules, we have $D^T = D$, $O^T = -O$. Notice that, if $\xi = i$, the operator in Eqs. (12) describes the real-time evolution according to a (pseudo)-Hamiltonian $\hat{A}$. On the other hand, if $\xi = 1$, it describes an imaginary-time evolution (related to the construction of the thermal ensemble at inverse temperature $\beta = 1/s$), that can be thought of as the analytic continuation of the purely imaginary evolution. As discussed in Section 4, in some particular cases, the action played by such kind of operator on a given state can be intended as part of a measurement process.

In what follows we discuss how to evaluate, for any (real) symmetric $\hat{A}$ operator, the $s$-evolved state

$$|\psi(s)\rangle = \hat{M}(s)|\psi\rangle,$$

taking as initial state $|\psi\rangle$ a generic Gaussian state of the form (2). In Appendix A we show that, because of the quadratic structure of $\hat{A}$, the state $|\psi(s)\rangle$ keeps the Gaussian form of Eq. (2):

$$|\psi(s)\rangle \propto \exp\left\{\frac{1}{2}\sum_{p,q} Z_{p,q}(s)\, \hat{c}_p^\dagger \hat{c}_q^\dagger\right\}|0\rangle_{\hat{c}}, \tag{13}$$

with

$$Z(s) = -\left[u^\dagger(s)\right]^{-1} v^\dagger(s), \tag{14}$$

and $u(s), v(s)$ submatrices of the $s$-evolved Bogoliubov matrix $\mathbb{U}(s)$. In the next Subsection we are going to show how to evaluate the matrix $\mathbb{U}(s)$ (an alternative derivation is provided in Appendix A).

### 3.1 Bogoliubov matrix evolution

To characterize the evolved state $|\psi(s)\rangle$, one only needs to know how the Bogoliubov matrix $\mathbb{U}(s)$ evolves with $\hat{M}(s)$. Exploiting the identity $\left[\hat{M}(s)\right]^{-1} = \hat{M}(-s)$, we can write

$$|\psi(s)\rangle = \hat{M}(s)|\psi\rangle \propto e^{\hat{\mathcal{Z}}(s)}\hat{M}(s)|0\rangle_{\hat{c}}\,, \tag{15}$$

where $|\psi\rangle$ is a generic Gaussian state given by Eq. (2). The operator

$$\hat{\mathcal{Z}}(s) = \frac{1}{2}\sum_{p,q} Z_{p,q}\,\hat{c}_p^{\dagger}(s)\,\hat{c}_q^{\dagger}(s)\,, \tag{16}$$

is defined through the evolved fermions

$$\hat{c}_q^{(\dagger)}(s) = \hat{M}(s)\,\hat{c}_q^{(\dagger)}\,\hat{M}(-s)\,, \tag{17}$$

obeying anticommutation relations $\{\hat{c}_q(s), \hat{c}_p^{\dagger}(s)\} = \delta_{p,q}$. Note that, for $\xi = 1$, the transformation $\hat{M}(s)$ is not unitary and the conjugation operation does not commute with the evolution one, hence $\hat{c}_q^{\dagger}(s) \neq \left[\hat{M}(s)\,\hat{c}_q\,\hat{M}(-s)\right]^{\dagger}$. Since $\hat{M}(s)|0\rangle_{\hat{c}} \propto |0\rangle_{\hat{c}}$, to determine the evolved state, one has to calculate $\hat{\mathcal{Z}}(s)$. This can be done by evaluating

$$\partial_s \hat{c}_q^{(\dagger)}(s) = \xi\,\hat{M}(s)\left[\hat{A}, \hat{c}_q^{(\dagger)}\right]\hat{M}(-s)\,. \tag{18}$$

Given the quadratic structure of the operator $\hat{A}$, the above commutators read

$$\left[\hat{A}, \hat{c}_q^{\dagger}\right] = \;\;2\sum_i \left(D_{i,q}\,\hat{c}_i^{\dagger} + O_{q,i}\,\hat{c}_i\right)\,, \tag{19a}$$

$$\left[\hat{A}, \hat{c}_q\right] = -2\sum_i \left(D_{i,q}\,\hat{c}_i + O_{q,i}\,\hat{c}_i^{\dagger}\right)\,. \tag{19b}$$

Therefore

$$\partial_s \hat{c}_q^{\dagger}(s) = \;\;2\xi\sum_i \left(O_{q,i}\,\hat{c}_i(s) + D_{i,q}\,\hat{c}_i^{\dagger}(s)\right)\,, \tag{20a}$$

$$\partial_s \hat{c}_q(s) = -2\xi\sum_i \left(D_{i,q}\,\hat{c}_i(s) + O_{q,i}\,\hat{c}_i^{\dagger}(s)\right)\,. \tag{20b}$$

At this point, we make the Ansatz that the fermionic operators depend on $s$ only through a Bogoliubov-like matrix

$$\hat{\Psi}(s) = \mathbb{U}^{\mathrm{ev}}(s)\hat{\Phi}\,, \quad \text{with} \quad \mathbb{U}^{\mathrm{ev}}(s) = \begin{pmatrix} u^{\mathrm{ev}}(s) & \bar{v}^{\mathrm{ev}}(s) \\ v^{\mathrm{ev}}(s) & \bar{u}^{\mathrm{ev}}(s) \end{pmatrix}\,. \tag{21}$$

The expression for the submatrices of the $\mathbb{U}^{\mathrm{ev}}(s)$ is given in Eq. (29). Note that, while for $\xi = i$ one has $\bar{u}^{\mathrm{ev}}(s) = [u^{\mathrm{ev}}(s)]^*$ and $\bar{v}^{\mathrm{ev}}(s) = [v^{\mathrm{ev}}(s)]^*$ (so that $\mathbb{U}^{\mathrm{ev}}(s)$ is a true Bogoliubov transformation), in general this is not true.

The above Ansatz is equivalent to assume that the $s$-evolved state $|\psi(s)\rangle$ is a Bogoliubov vacuum $\hat{\gamma}_{\mu}(s)|\psi(s)\rangle = 0,\ \forall\mu$. By substituting this Ansatz in Eq. (20), we obtain a set of differential equations for the elements of the evolved matrix:

$$\partial_s u_{q,\mu}^{\mathrm{ev}}(s) = -2\xi\sum_i \left(D_{q,i}\,u_{i,\mu}^{\mathrm{ev}}(s) + O_{q,i}\,v_{i,\mu}^{\mathrm{ev}}(s)\right)\,, \tag{22a}$$

$$\partial_s \bar{v}_{q,\mu}^{\mathrm{ev}}(s) = -2\xi\sum_i \left(D_{q,i}\,\bar{v}_{i,\mu}^{\mathrm{ev}}(s) + O_{q,i}\,\bar{u}_{i,\mu}^{\mathrm{ev}}(s)\right)\,, \tag{22b}$$

$$\partial_s v_{q,\mu}^{\mathrm{ev}}(s) = \;\;2\xi\sum_i \left(D_{q,i}\,v_{i,\mu}^{\mathrm{ev}}(s) + O_{q,i}\,u_{i,\mu}^{\mathrm{ev}}(s)\right)\,, \tag{22c}$$

$$\partial_s \bar{u}_{q,\mu}^{\mathrm{ev}}(s) = \;\;2\xi\sum_i \left(D_{q,i}\,\bar{u}_{i,\mu}^{\mathrm{ev}}(s) + O_{q,i}\,\bar{v}_{i,\mu}^{\mathrm{ev}}(s)\right)\,, \tag{22d}$$

with the initial condition: $u_{q,\mu}^{\text{ev}}(0) = [\bar{u}_{q,\mu}^{\text{ev}}(0)]^* = u_{q,\mu}$, and $v_{q,\mu}^{\text{ev}}(0) = [\bar{v}_{q,\mu}^{\text{ev}}(0)]^* = v_{q,\mu}$. In a compact form, these equations read

$$\partial_s \mathbb{U}^{\text{ev}}(s) = -2\,\xi\,\mathbb{A}\,\mathbb{U}^{\text{ev}}(s), \quad \text{with} \quad \mathbb{A} = \begin{pmatrix} D & O \\ -O & -D \end{pmatrix}, \tag{23}$$

being $D^T = D$, $O^T = -O$. This leads to the formal solution

$$\mathbb{U}^{\text{ev}}(s) = e^{-2\xi\mathbb{A}s}\,\mathbb{U}\,.^2 \tag{24}$$

The matrix $\mathbb{U}^{\text{ev}}(s)$ is not *a priori* characterizing a fermionic state in that it should be unitary, to guarantee the normalization of the state, and should obey Eqs. (4), to impose the anticommutation relations for the $\hat{\Psi}(s)$. While the former requirement is automatically fulfilled for $\xi = i$ (because of the unitarity of the evolution operator), in the case $\xi = 1$ it must be forced by performing a QR-decomposition, as discussed in details in the next subsection. Once the unitarity of the matrix is imposed, Eqs. (4) are satisfied if the Bogoliubov matrix preserves the symplectic structure of Eq. (6), and this happens for $\hat{A}$ being Hermitian.

By substituting the normalized s-evolved Bogoliubov matrix in $Z(s)$ [cf. Eq. (14)], we obtain the expression for $|\psi(s)\rangle$. If the above conditions are satisfied, this is a Gaussian fermionic state, as the one in Eq. (13). A more detailed discussion about this point is provided in Appendix A.

## 3.2 QR decomposition

We now discuss how to restore the unitarity of the s-evolved Bogoliubov matrix $\mathbb{U}^{\text{ev}}(s)$, when considering a norm-non preserving evolution with $\xi = 1$, that is in general obtained by QR-decomposing the s-evolved Bogoliubov matrix $\mathbb{U}^{\text{ev}}(s)$. This procedure keeps the $Z(s)$ matrix in Eq. (14) – and then the Gaussian state – invariant, as we are going to see below. Nevertheless, it is necessary to apply the known formulae [78] for evaluating the entanglement entropy and the local observables.

The QR-decomposition is a procedure that allows one to decompose a generic matrix as $K = U_Q R$, with $U_Q$ a unitary matrix and $R$ an upper triangular one. In what follows, we provide an argument to show that, if $\hat{A} = \hat{A}^\dagger$, the unitary matrix obtained by QR-decomposing the s-evolved Bogoliubov matrix $\mathbb{U}^{\text{ev}}(s)$ describes a fermionic state, i.e., it has the symplectic form of Eq. (6). The straightforward way would be to check it by construction, a procedure that requires some involved calculations. Here we choose a different strategy. Since $R$ is positive definite (we have checked it numerically), the QR decomposition must be unique. Therefore, we assume $U_Q$ to have the right structure and then we show that it exists a decomposition that is compatible with this assumption.

Let us consider a symmetric operator $\hat{A}$ defined in Eq. (12b) and the associated coefficient matrix $\mathbb{A}$ defined in Eq. (23). For simplicity let us assume $\hat{A}$ to have nearest-neighbors interactions, i.e. $D_{i,j} = O_{i,j} = 0$ if $j \neq i + 1$, and let as assume $D_{i,i+1} = O_{i,i+1}$ (e.g. the Kitaev Hamiltonian). The most general case does not have an analytical solution, but the main argument of this section should hold independently. Because of the symmetries of $D$ and $O$, we have that $\{D, O\} = 0$.[3] We notice that

$$\mathbb{A}^{2n} = \begin{pmatrix} X^{2n} & 0 \\ 0 & X^{2n} \end{pmatrix}, \quad \text{with} \quad X^2 = D^2 - O^2. \tag{25}$$

---

[2]For $s = it$ and $\hat{A} = \hat{H}$, it returns the usual Hamiltonian time evolution of the system [78].

[3]This simply follows from the fact that $(\{D, O\})^\dagger = -\{D, O\}$. Being $D$ and $O$ real matrices, it implies that the anticommutator vanishes.

By exploiting the block diagonal form of Eq. (25) and the identities

$$\sum_{n=0}^{\infty} \frac{x^{2n}}{(2n)!} = \cosh(x), \qquad \sum_{n=0}^{\infty} \frac{x^{2n}}{(2n+1)!} = \frac{\sinh(x)}{x}, \tag{26}$$

we can write

$$e^{-2\xi \mathbb{A}s} = \sum_{n=0}^{\infty} \left[ \frac{(-2\xi \mathbb{A}s)^{2n}}{(2n)!} + \frac{-2\xi \mathbb{A}s \,(-2\xi \mathbb{A}s)^{2n}}{(2n+1)!} \right] = \begin{pmatrix} \mathrm{ch}^{+}(s) & \mathrm{sh}(s) \\ -\mathrm{sh}(s) & \mathrm{ch}^{-}(s) \end{pmatrix}, \tag{27}$$

where, for shortness of notation, we have posed

$$\mathrm{ch}^{\pm}(s) = I \cosh(-2\xi X s) \pm D \sinh(-2\xi X s) X^{-1}, \qquad \mathrm{sh}(s) = O \sinh(-2\xi X s) X^{-1}. \tag{28}$$

Notice that $[\mathrm{ch}^{\pm}(s)]^{\dagger} = \mathrm{ch}^{\pm}(s)$, while $\mathrm{sh}^{\dagger}(s) = -\mathrm{sh}(s)$. Moreover, since $\{D, O\} = 0$, we have $[D, \mathrm{ch}^{\pm}(s)] = [O, \mathrm{ch}^{\pm}(s)] = 0$ and $[D, \mathrm{sh}(s)] = [O, \mathrm{sh}(s)] = 0$. This means that

$$\begin{pmatrix} u^{\mathrm{ev}}(s) & \bar{v}^{\mathrm{ev}}(s) \\ v^{\mathrm{ev}}(s) & \bar{u}^{\mathrm{ev}}(s) \end{pmatrix} = \begin{pmatrix} \mathrm{ch}^{+}(s)u + \mathrm{sh}(s)v & \mathrm{ch}^{+}(s)v^{*} + \mathrm{sh}(s)u^{*} \\ -\mathrm{sh}(s)u + \mathrm{ch}^{-}(s)v & -\mathrm{sh}(s)v^{*} + \mathrm{ch}^{-}(s)u^{*} \end{pmatrix}. \tag{29}$$

We can QR-decompose $\mathbb{U}^{\mathrm{ev}}(s)$

$$\mathbb{U}^{\mathrm{ev}}(s) = \mathbb{U}(s)\,\mathrm{R}(s), \tag{30}$$

making the Ansatz that the $\mathbb{U}(s)$ preserves the symplectic form of Eq. (6), so that (30) can be rewritten as

$$\begin{pmatrix} u^{\mathrm{ev}}(s) & \bar{v}^{\mathrm{ev}}(s) \\ v^{\mathrm{ev}}(s) & \bar{u}^{\mathrm{ev}}(s) \end{pmatrix} = \begin{pmatrix} u(s) & v^{*}(s) \\ v(s) & u^{*}(s) \end{pmatrix} \begin{pmatrix} r_1(s) & r_3(s) \\ 0 & r_2(s) \end{pmatrix}. \tag{31}$$

In what follows, we show that there exists a decomposition compatible with this Ansatz, by imposing Eq. (4) and that, because of the unicity, it must be the only possible. Let us assume to have derived $r_1(s)$ by construction. We can explicitly write $\mathbb{U}(s) = \mathbb{U}^{\mathrm{ev}}(s)R(s)^{-1}$, exploiting the fact that the inverse of an upper block triangular matrix remains an upper block triangular matrix:

$$\begin{pmatrix} u(s) & v^{*}(s) \\ v(s) & u^{*}(s) \end{pmatrix} = \begin{pmatrix} u^{\mathrm{ev}}(s)\,r_1^{-1}(s) & u^{\mathrm{ev}}(s)\,r_3^{-1}(s) + \bar{v}^{\mathrm{ev}}(s)\,r_2^{-1}(s) \\ v^{\mathrm{ev}}(s)\,r_1^{-1}(s) & v^{\mathrm{ev}}(s)\,r_3^{-1}(s) + \bar{v}^{\mathrm{ev}}(s)\,r_2^{-1}(s) \end{pmatrix}. \tag{32}$$

We can substitute these expressions in Eq. (4) to obtain

$$u^{\dagger}u + v^{\dagger}v = \left[r_2^{T}(s)\right]^{-1} \left([\bar{u}^{\mathrm{ev}}]^{T}(s)u^{\mathrm{ev}}(s) + [\bar{v}^{\mathrm{ev}}]^{T}(s)v^{\mathrm{ev}}(s)\right)\left[r_1(s)\right]^{-1} = I. \tag{33}$$

Since $[\bar{u}^{\mathrm{ev}}]^{T}(s)u^{\mathrm{ev}}(s) + [\bar{v}^{\mathrm{ev}}]^{T}(s)v^{\mathrm{ev}}(s) = I$, the above equation provides a relation between two of the blocks of the matrix $R(s)$:

$$r_2(s) = \left[r_1^{T}(s)\right]^{-1}. \tag{34}$$

By imposing the second constraint in Eq. (4), we find the relation for $r_3(s)$

$$r_3(s) = -r_1^{T}\left([u^{\mathrm{ev}}]^{\dagger}(s)\bar{v}^{\mathrm{ev}}(s) + [v^{\mathrm{ev}}]^{\dagger}(s)\bar{u}^{\mathrm{ev}}(s)\right)^{-1}\left([u^{\mathrm{ev}}]^{\dagger}(s)u^{\mathrm{ev}}(s) + [v^{\mathrm{ev}}]^{\dagger}(s)v^{\mathrm{ev}}(s)\right). \tag{35}$$

As a last comment we notice that the matrix $Z(s)$ in Eq. (14) –and then the Gaussian state– are left invariant by the application of the QR decomposition:

$$Z(s) = \left[u^{\dagger}(s)\right]^{-1}v(s)^{\dagger} = \left\{\left[u^{\mathrm{ev}}(s)\,r_1^{-1}(s)\right]^{\dagger}\right\}^{-1}\left[v^{\mathrm{ev}}(s)\,r_1^{-1}(s)\right]^{\dagger} = \left\{[u^{\mathrm{ev}}]^{\dagger}(s)\right\}^{-1}[v^{\mathrm{ev}}]^{\dagger}(s). \tag{36}$$

# 4  Number-preserving operators

We now consider a generic number-preserving operator

$$\hat{A}_{\mathcal{I},\mathcal{J}} = \sum_{i\in\mathcal{I}} \sum_{j\in\mathcal{J}} \hat{c}_i^\dagger \hat{c}_j \,, \tag{37}$$

where $\mathcal{I}, \mathcal{J}$ are two sets of indices labeling certain sites on the system, and apply

$$\hat{M}_{\mathcal{I},\mathcal{J}} = e^{\beta \hat{A}_{\mathcal{I},\mathcal{J}}} \,, \tag{38}$$

to a Gaussian state. We focus on this particular example of the theory presented in Section 3.1, since it generalizes the construction discussed in Ref. [49, 50]. By exploiting the anticommutation relations, it is possible to show that $\hat{A}_{\mathcal{I},\mathcal{J}}^n = \alpha^{n-1} \hat{A}_{\mathcal{I},\mathcal{J}}$, with $\alpha = |\mathcal{I} \cap \mathcal{J}|$. In fact

$$\hat{A}_{\mathcal{I},\mathcal{J}}^2 = \sum_{i,l\in\mathcal{I}} \sum_{j,m\in\mathcal{J}} \hat{c}_i^\dagger \hat{c}_j \hat{c}_l^\dagger \hat{c}_m = \sum_{i,l\in\mathcal{I}} \sum_{j,m\in\mathcal{J}} \left( \hat{c}_i^\dagger \hat{c}_j \delta_{l,m} + \hat{c}_i^\dagger \hat{c}_j \hat{c}_m \hat{c}_l^\dagger \right)$$

$$= \sum_{i,l\in\mathcal{I}} \sum_{j,m\in\mathcal{J}} \left[ \hat{c}_i^\dagger \hat{c}_j \delta_{l,m} + \tfrac{1}{2} \hat{c}_i^\dagger (\hat{c}_j \hat{c}_m + \hat{c}_m \hat{c}_j) \hat{c}_l^\dagger \right] = |\mathcal{I} \cap \mathcal{J}| \sum_{i\in\mathcal{I}} \sum_{j\in\mathcal{J}} \hat{c}_i^\dagger \hat{c}_j \,, \tag{39}$$

where the last equality holds, since $j$ and $m$ run in the same set. Therefore, if $\mathcal{I} \cap \mathcal{J} = \varnothing$, we have $\hat{A}_{\mathcal{I},\mathcal{J}}^2 = 0$ and $\hat{M}_{\mathcal{I},\mathcal{J}} = \hat{1} + \beta \hat{A}_{\mathcal{I},\mathcal{J}}$, where $\hat{1}$ denotes the identity operator. Otherwise

$$\hat{M}_{\mathcal{I},\mathcal{J}} = \sum_{n=0}^\infty \frac{(\beta \hat{A}_{\mathcal{I},\mathcal{J}})^n}{n!} = \hat{1} + \frac{(e^{\alpha\beta} - 1)\hat{A}_{\mathcal{I},\mathcal{J}}}{\alpha} \,. \tag{40}$$

In what follows, without loss of generality, we set $\beta \equiv \log(\alpha + 1)/\alpha$, so that $\hat{M}_{\mathcal{I},\mathcal{J}} = \hat{1} + \hat{A}_{\mathcal{I},\mathcal{J}}$ reduces to that studied in Ref. [49, 50] for $\mathcal{I} = \mathcal{J} = \{i\}$. As discussed in Appendix B, the application of such a $\hat{M}_{\mathcal{I},\mathcal{J}}$ can be thought as the measurement step of a quantum-jump dynamics.

We want to evaluate the effect of this operator on a Gaussian state

$$|\psi\rangle_{\hat{M}_{\mathcal{I},\mathcal{J}}} \equiv \hat{M}_{\mathcal{I},\mathcal{J}} |\psi\rangle = (\hat{1} + \hat{A}_{\mathcal{I},\mathcal{J}}) |\psi\rangle \,. \tag{41}$$

Following the procedure discussed in Section 3.1, we write the evolved state as

$$|\psi\rangle_{\hat{M}_{\mathcal{I},\mathcal{J}}} \propto \exp\left\{ \frac{1}{2} \sum_{p,q} Z_{p,q} \hat{k}_p^\dagger \hat{k}_q^\dagger \right\} |0\rangle \,, \tag{42}$$

where we introduced the transformed fermionic operators $\hat{k}_q^{(\dagger)} = \hat{M}_{\mathcal{I},\mathcal{J}} \hat{c}_q^{(\dagger)} \hat{M}_{\mathcal{I},\mathcal{J}}^{-1}$. By exploiting the fact that $\hat{M}_{\mathcal{I},\mathcal{J}}^{-1} = \hat{1} - (\alpha + 1)^{-1} \hat{A}_{\mathcal{I},\mathcal{J}}$, we derive the exact expression of $\hat{k}^{(\dagger)}$

$$\hat{k}_q^\dagger = \hat{c}_q^\dagger + \sum_{j\in\mathcal{J}} \delta_{q,j} \sum_{i\in\mathcal{I}} \hat{c}_i^\dagger \,, \tag{43a}$$

$$\hat{k}_q = \hat{c}_q - (\alpha + 1)^{-1} \sum_{j\in\mathcal{J}} \delta_{q,j} \sum_{i\in\mathcal{I}} \hat{c}_i \,. \tag{43b}$$

This leads to the following expressions for the evolved Bogoliubov matrix:

$$u_{q,\mu}^{\mathrm{ev}} = u_{q,\mu} - (\alpha + 1)^{-1} \sum_{j\in\mathcal{J}} \delta_{q,j} \sum_{i\in\mathcal{I}} u_{i,\mu} \,, \qquad \bar{u}_{q,\mu}^{\mathrm{ev}} = u_{q,\mu}^* + \sum_{j\in\mathcal{J}} \delta_{q,j} \sum_{i\in\mathcal{I}} u_{i,\mu}^* \,, \tag{44a}$$

$$\bar{v}_{q,\mu}^{\mathrm{ev}} = v_{q,\mu}^* - (\alpha + 1)^{-1} \sum_{j\in\mathcal{J}} \delta_{q,j} \sum_{i\in\mathcal{I}} v_{i,\mu}^* \,, \qquad v_{q,\mu}^{\mathrm{ev}} = v_{q,\mu} + \sum_{j\in\mathcal{J}} \delta_{q,j} \sum_{i\in\mathcal{I}} v_{i,\mu} \,. \tag{44b}$$

As discussed in Appendix C, an alternative derivation can be obtained from the evolved antisymmetric matrix $Z$.

## 4.1 Local operator

The above results can be applied straightforwardly to local (l) operators, as for the one used in Ref. [49,50], $\hat{A}^l_{\mathcal{I},\mathcal{J}} \equiv \hat{n}_i = \hat{c}^\dagger_i \hat{c}_i$, such that $\hat{M}^l_{\mathcal{I},\mathcal{J}} \equiv e^{\log(2)\hat{A}^l} = \hat{1} + \hat{c}^\dagger_i \hat{c}_i$. The Bogoliubov matrix evolves according to Eqs. (44), which read

$$u^{ev,l}_{q,\mu} = (1 - \delta_{q,i}/2)u_{q,\mu}, \qquad \bar{u}^{ev,l}_{q,\mu} = (1 + \delta_{q,i})u^*_{q,\mu}, \qquad (45a)$$

$$\bar{v}^{ev,l}_{q,\mu} = (1 - \delta_{q,i}/2)v^*_{q,\mu}, \qquad v^{ev,l}_{q,\mu} = (1 + \delta_{q,i})v_{q,\mu}. \qquad (45b)$$

This evolution is implemented by applying to the $\mathbb{U}$ a block-diagonal matrix T with entries $T_{i+L,i+L} = T^{-1}_{i,i} = 2$ (see Eq. (1) of Ref. [50]).

## 4.2 Two-site operator

We now consider a non-local (nl) operator with support on two sites ($\mathcal{I} = \mathcal{J} = \{i, j\}$):

$$\hat{A}^{nl}_{\mathcal{I},\mathcal{J}} = \left(\hat{c}^\dagger_i + \hat{c}^\dagger_j\right)\left(\hat{c}_i + \hat{c}_j\right). \qquad (46)$$

Since $\alpha = |\mathcal{I} \cap \mathcal{J}| = 2$, we have $\hat{M}^{nl}_{\mathcal{I},\mathcal{J}} \equiv e^{\frac{\log(3)}{2}\hat{A}^{nl}} = \hat{1} + \hat{A}^{nl}_{\mathcal{I},\mathcal{J}}$. Without losing in generality, in the following we will focus on one dimensional systems and set $j = i + r$ (with $r > 0$). As we shall see in the next Section, a measurement operator as the one in Eq. (46) may unveil a richer physics than local ones. In fact, in the spin-1/2 language, it corresponds to a string operator. The evolved Bogoliubov matrix can be derived from Eqs. (44) which, in this case, read

$$u^{ev,nl}_{q,\mu} = u_{q,\mu} - \frac{1}{3}(\delta_{q,i} + \delta_{q,i+r})(u_{i,\mu} + u_{i+r,\mu}), \qquad (47a)$$

$$\bar{v}^{ev,nl}_{q,\mu} = v^*_{q,\mu} - \frac{1}{3}(\delta_{q,i} + \delta_{q,i+r})(v^*_{i,\mu} + v^*_{i+r,\mu}), \qquad (47b)$$

$$\bar{u}^{ev,nl}_{q,\mu} = u^*_{q,\mu} + (\delta_{q,i} + \delta_{q,i+r})(u^*_{i,\mu} + u^*_{i+r,\mu}), \qquad (47c)$$

$$v^{ev,nl}_{q,\mu} = v_{q,\mu} + (\delta_{q,i} + \delta_{q,i+r})(v_{i,\mu} + v_{i+r,\mu}). \qquad (47d)$$

# 5 Quantum jumps with string operators

In this section we use the machinery presented above to describe the evolution of a quantum Ising chain under the effect of a quantum-jump dynamics generated by $\hat{M}^{nl}_{\mathcal{I},\mathcal{J}}$ [cf. Eq. (46) and below], with emphasis on the evolution of the entanglement entropy.

We consider the Hamiltonian

$$\hat{H} = -J\sum_j \hat{\sigma}^x_j \hat{\sigma}^x_{j+1} - h\sum_j \hat{\sigma}^z_j, \qquad (48)$$

where $\hat{\sigma}^\alpha_j$ ($\alpha = x, y, z$) are the usual spin-1/2 Pauli matrices acting on the $j$-th site, while $J > 0$ denotes the ferromagnetic spin-spin coupling and $h$ the transverse magnetic field. In the thermodynamic limit, the ground state exhibits a zero-temperature continuous transition at $|h/J| = 1$, separating a paramagnetic (symmetric) phase, for $|h/J| > 1$, from a ferromagnetic (symmetry-broken) phase, for $|h/J| < 1$ [79,80]. Through a Jordan-Wigner transformation [81,82],

$$\hat{\sigma}^z_j = 1 - 2\hat{n}_j, \quad \hat{\sigma}^+_j = \hat{K}_j \hat{c}_j, \quad \text{with} \quad \hat{K}_j = \Pi^{j-1}_{j'}(1 - 2\hat{n}_{j'}), \qquad (49)$$

the Ising model (48) can be mapped into the so-called Kitaev chain

$$\hat{H} = -J\sum_j \left(\hat{c}^\dagger_j \hat{c}_{j+1} + \hat{c}^\dagger_j \hat{c}^\dagger_{j+1} + \text{h.c.}\right) - 2h\sum_j \hat{c}^\dagger_j \hat{c}_j. \qquad (50)$$

This Hamiltonian is of the form (1), with

$$Q_{j,j} = h, \qquad Q_{j,j+1} = Q_{j+1,j} = -J/2, \qquad P_{j,j+1} = -P_{j+1,j} = -J/2. \tag{51}$$

Hereafter we work in units of $\hbar = 1$, set $J = 1$ as the energy scale, and assume periodic boundary conditions for fermions.

We implement the quantum-jump protocol described in Appendix B on the fermionic model above, by initializing the system in the ground state of the Hamiltonian (50) with a given value of $h$ and then Trotterizing the time evolution in steps $\delta t \propto (4L\gamma)^{-1}$, to ensure convergence, being $\gamma$ the strength of the measurements. The emerging dynamics is one of the possible unravelings [10] of the Lindblad master equation [9]

$$\dot{\rho}(t) = -i[\hat{H}, \rho(t)] - \gamma \sum_j \left( \hat{A}_j(r)\rho(t)\hat{A}_j(r) - \tfrac{1}{2}\{\hat{A}_j^2(r), \rho(t)\} \right), \tag{52}$$

being $\hat{A}_j(r) = (\hat{c}_j^\dagger + \hat{c}_{j+r}^\dagger)(\hat{c}_j + \hat{c}_{j+r})$ the same operator as in Eq. (46), with $i \equiv j + r$.[4] In the spin language, this operator is a sum of strings of Pauli matrices, of length $r$:

$$\hat{A}_j(r) = \hat{1} + \hat{\sigma}_j^- (\hat{K}_j \hat{K}_{j+r}) \hat{\sigma}_{j+r}^+ + \hat{\sigma}_j^+ (\hat{K}_j \hat{K}_{j+r}) \hat{\sigma}_{j+r}^- - (\hat{\sigma}_j^z + \hat{\sigma}_{j+r}^z)/2, \tag{53}$$

with $\hat{\sigma}_j^\pm = (\hat{\sigma}_j^x \pm i\hat{\sigma}_j^y)/2$. Hereafter, we refer to it as the string operator.

We point out that our framework is similar to the one presented in Ref. [42], although with some important qualitative differences. The former is described by a random sequence of stabilizer-formalism measurements of Pauli string operators. These operators have a fixed range $r$ and are taken from a random distribution of strings, located around different sites, and made of different sets of onsite Pauli operators.

In this work, in contrast, we are following an unraveling of the quantum master equation Eq. (52). As detailed in Appendix B, we apply random quantum-jump measurements of string operators $\hat{1} + \hat{A}_j(r)$, located at different sites, but with the structure fixed as the one in Eq. (53). This is sufficient to achieve a measurement-induced nontrivial dynamics, with important consequences on the entanglement generation.

It is important to emphasize that the operators $\hat{1} + \hat{A}_j(r)$ have the exponential form given by Eqs. (38) and (40) (see also Sec. 4). In this way, the state keeps its Gaussian form under the application of the quantum jumps, and a numerical analysis of quite large system sizes is possible.

## 5.1 Entanglement entropy

Let us focus our attention on the entanglement entropy. Given a pure state $|\psi\rangle$, there is an unambiguous way to quantify its quantum correlations. We divide the system in two sub-chains, one with length $\ell$ and the other with length $L - \ell$. After evaluating the density matrix $\rho_\ell = \text{Tr}_{L-\ell}[|\psi\rangle\langle\psi|]$ reduced to the subsystem $\ell$, the entanglement entropy is quantified by its von Neumann entropy [7,8]:

$$S_\ell[\rho_\ell] = -\rho_\ell \log \rho_\ell. \tag{54}$$

For the Ising model we are discussing herewith, it is known that after, a sudden quench in a purely Hamiltonian dynamics, the entanglement entropy approaches a stationary value that increases linearly with $\ell$, thus obeying a volume-law behavior [3,4]. As discussed in Refs. [44–51], this dynamical volume law is generally destroyed by any arbitrary small local

---

[4]Note that, with our convention for the boundary conditions, we have $\hat{c}_{j+r}^{(\dagger)} \equiv \hat{c}_{j+r-L}^{(\dagger)}$ for any $j + r > L$.

weak measurement process. In this scenario, the asymptotic entanglement entropy is conjectured to exhibit either a logarithmic scaling or an area-law behavior, depending on the strength of measurements and of the Hamiltonian parameters. We are going to show that the situation is quite different for the case of the nonlocal operators defined in Eq. (53).

In practice, to avoid spurious contributions from classical correlations, we evaluate Eq. (54) on each stochastic quantum trajectory $|\psi_t\rangle$ (defined in Appendix B), and then average over different stochastic realizations.[5] For simplicity of notation, we denote the ensemble-averaged entanglement entropy as

$$S_\ell(t) \equiv \left\langle\!\!\left\langle S_\ell\big[\rho_\ell^{\text{stoch}}(t)\big]\right\rangle\!\!\right\rangle, \quad \text{where} \quad \rho_\ell^{\text{stoch}}(t) = \text{Tr}_{L-\ell}\big[|\psi_t\rangle\langle\psi_t|\big], \tag{55}$$

and where the symbol $\left\langle\!\!\left\langle \cdots \right\rangle\!\!\right\rangle$ denotes the average over different stochastic realizations. We also define the asymptotic entanglement entropy as

$$\overline{S}_\ell = \lim_{T\to\infty} \int_{t^\star}^{T} dt' S_\ell(t').^{6} \tag{56}$$

We remark that, in all the simulations shown hereafter, we keep the size of the partition fixed to $\ell = L/4$ and study the scaling with $L$.

Below we present two different scenarios for fixed $\gamma = 0.5$ and for $r = 1$, 4, $L/2$: the measurement-only dynamics (Section 5.2) and the full quantum-jump dynamics (Section 5.3). Finally, in Section 5.4, we discuss how these results may change when varying the measurement rate $\gamma$, as well as the range $r$ and the partition size $\ell$.

## 5.2 Measurement-only dynamics

We first neglect the Hamiltonian contribution, i.e., we implement the quantum-jump protocol described in Appendix B with $\hat{H} = 0$. The Hamiltonian is only used to set the initial state as its corresponding ground state with $h = 0.5$. However, we have carefully checked that the choice of the initial state does not affect the asymptotic value reached by $S_\ell(t)$, but only its transient dynamics.[7] The advantage of studying a measurement-only dynamics is to isolate the effects of measuring the string operators, without considering the interplay with the unitary dynamics.

The three upper panels of Figure 1 report the ensemble-averaged entanglement entropy $S_\ell(t)$ versus the rescaled time $\gamma t$, for three ranges of the string measurement operator: $r = 1$ [panel (a)], $r = 4$ [panel (b)], and $r = L/2$ [panel (c)], and for different system sizes $L$. We note that, for any finite value of $r$ (i.e., which does not depend on $L$), the entanglement entropy displays metastable plateaus, whose lifetime generally increases with some power of $L$ (see the discussion below, following the rescaled data of Fig. 3). This may cause some difficulties in estimating the long-time asymptotic behavior, especially for large system size. However we checked that the qualitative scaling of the entanglement entropy with $L$ does not change if one considers the value reached in one of such steady values. Moreover, since their lifetime grows with the size, in the thermodynamic limit we can treat the first emerging plateau as the effective stationary state.

Going into the details, for $r = 1$ we observe that the measurement-only dynamics generally destroys entanglement in time and the system stabilizes on a low-entangled asymptotic state

---

[5]For details on how to evaluate the von Neumann entropy of the reduced density matrix of Gaussian states see, e.g., Ref. [78].

[6]Ensemble-averaged entanglement entropies have then been obtained with $N_{\text{avg}} = 10^2$ different quantum trajectories. Subsequently, to perform time averages, we fix a given evolution time $T$ and average in the interval $t' \in [0.7\,T, T]$.

[7]Note that the perfectly paramagnetic state with spins pointing along the $z$ direction ($h = \infty$) is a dark state of the dissipation, therefore, initializing the system close to such state leads to slower convergence.

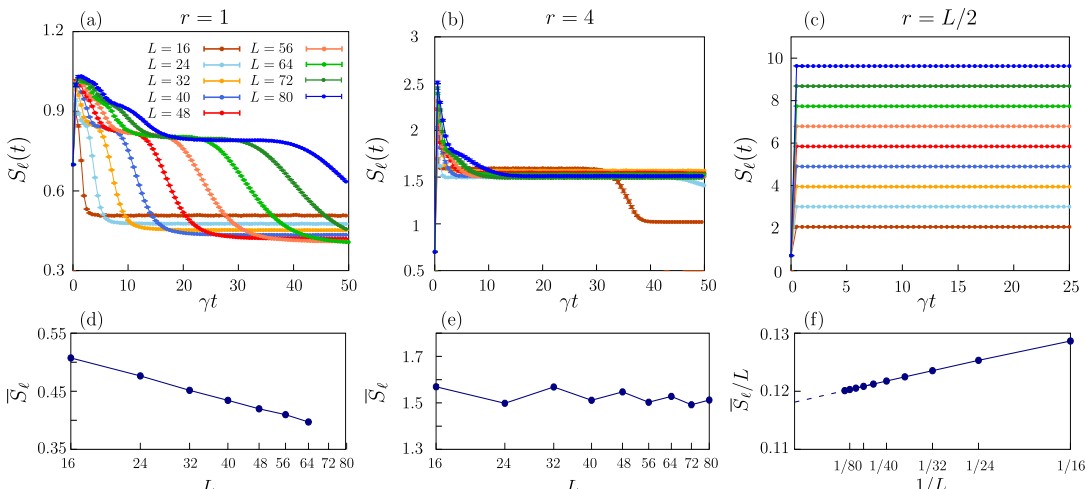

Figure 1: Top panels: the ensemble-averaged entanglement entropy $S_\ell(t)$ versus the rescaled time $\gamma t$, along the quantum-jump process with $\hat{H} = 0$. The different panels refer to $r = 1, 4, L/2$ [resp. (a), (b), (c)]. The various curves are for different sizes $L$ [see color code in (a)]. Bottom panels: scaling of the long-time asymptotic value $\overline{S}_\ell$ versus $L$, for $r = 1, 4$, and $\overline{S}_\ell/L$ versus $1/L$, for $r = L/2$ [resp. (d), (e), and (f)]. The dashed line in (f) extrapolates to the asymptotic value for $L \to \infty$.

[Fig. 1(a)]. This is confirmed by the data reported in Fig. 1(d), which shows the behavior of the asymptotic entanglement entropy $\overline{S}_\ell$ with the system size, as extrapolated by the second plateau stabilizing around the rescaled time $\gamma t \gtrsim 10^{-2} L^2$ [see also Fig. 3(a)]. A decreasing behavior in $L$ is clearly visible: $\overline{S}_\ell$ should saturate to a constant value for small enough ratio $r/L$, i.e., in the limit of local measurements, thus obeying an area-law behavior, as expected by a comparison with the case where onsite operators are measured [49, 50].

When the correlation range of the dissipative string operator is increased, as for $r = 4$ [Fig. 1(b)], the measurements still produce a low entangled state. In fact, the asymptotic value of the entanglement entropy approaches a constant value with $L$, with superimposed staggered oscillations, probably due to finite-size commensurability effects between $r$, $\ell$, $L$ [Fig. 1(e)]. It is worth noticing that, also in this case, we extrapolated the asymptotic value $\overline{S}_\ell$ over the first observable metastable plateau itself. When going to even later times, we find that the subsequent plateau, corresponding to a lower value of the entropy, is still obeying an area-law scaling (at least for $L \leq 48$, so that we were able to reach such plateau).

The above scenario dramatically changes when $r$ is chosen in such a way to be a thermodynamic fraction of $L$ [see Fig. 1(c), for $r = L/2$]. In that case, the measurement-only dynamics is projecting over Bell pairs at arbitrary distances, thus quickly leading the bipartite entanglement to an asymptotic stationary value which undergoes a volume-law behavior. This is clarified by Fig. 1(f), which shows $\overline{S}_\ell/L$ versus $L^{-1}$: the saturation of $\overline{S}_\ell/L$ to a finite value for $L^{-1} \to 0$ (see the dashed line, which extrapolates to the asymptotic value in the thermodynamic limit) clearly evidences a linear growth $\overline{S}_\ell \propto L$ for large system sizes.

## 5.3 Interplay with the Hamiltonian dynamics

We now switch on the Kitaev Hamiltonian $\hat{H}$ and follow the full quantum-jump dynamics for $h = 0.1$, starting from the ground state with $h_i = 0.5$. As for the case with $\hat{H} = 0$, we have checked that the asymptotic values reached by the entanglement entropy are independent of $h_i$, while they may depend on $h$ (although the qualitative behaviors with $L$ are unaffected by this choice).

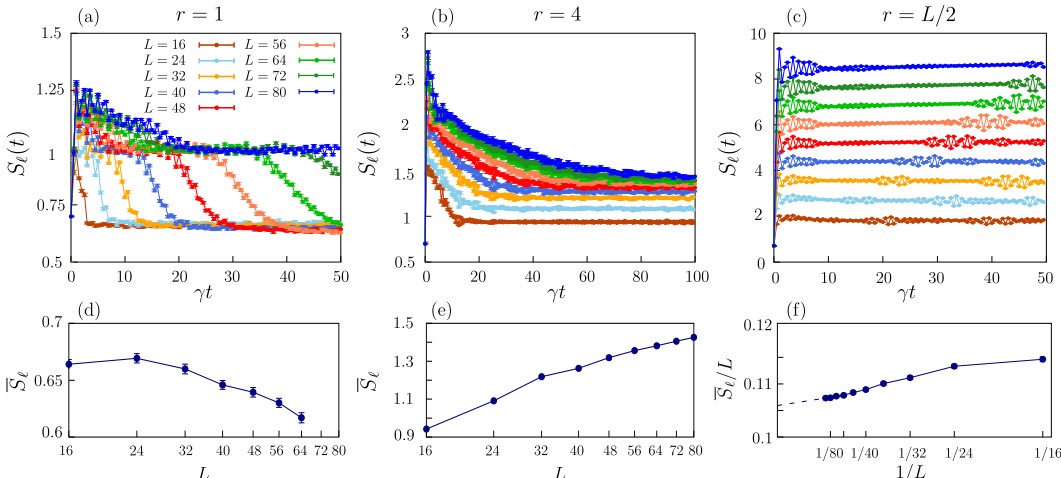

Figure 2: Same plots of Figure 1, but in the simultaneous presence of the Kitaev Hamiltonian $\hat{H}$ and of the string measurement process. We observe qualitatively similar behaviors, although with different time scales. Here we fix $\gamma = 0.5$, $J = 1$, $h = 0.1$, and start from the ground state of $\hat{H}$ with $h_i = 0.5$.

Figure 2 displays $S_\ell(t)$ versus $\gamma t$ (upper panels) and the asymptotic value $\overline{S}_\ell$ versus $L$, for the three ranges of $r = 1$ [panels (a),(d)], $r = 4$ [panels (b),(e)], and $r = L/2$ [panels (c),(f)]. The data shown have been obtained for $\gamma = 0.5$. In Fig. 2(a) we focus on the local case $r = 1$. The long-time averaged entanglement entropy attains a stationary value that, as for the measurement-only case, slightly decreases with the system size, thus reflecting an area-law behavior [see Fig. 2(d), which shows $\overline{S}_\ell$ corresponding to the plateau reached after a transient time $\gamma t \gtrsim 10^{-2}L^2$, as visible in Fig. 3(c)]. The value of the asymptotic entanglement entropy is slightly larger than the one obtained for the measurement-only dynamics, without the entangling action of $\hat{H}$ [compare with Fig. 1(a)].

Figure 2(b) reports the case $r = 4$, where we observe convergence in time to a single plateau, up to the times we were able to achieve numerically. Comparing the time behavior of the various curves with the corresponding ones for $\hat{H} = 0$ [see Fig. 1(b)], it is reasonable to associate this only plateau with the late-time second plateau emerging with $\hat{H} = 0$. In the presence of the Hamiltonian dynamics, the transient time appears to be reduced, with respect to that for the measurement-only dynamics. The asymptotic data shown in Fig. 2(e) ($x$-axis in logarithmic scale) suggest that $\overline{S}_\ell$ may grow sub-extensively with $L$, for small system sizes, while we expect it to eventually attain an area-law regime for large enough $L$. Physically, this can be understood by the fact that, for $L$ much larger than $r$, the dissipation becomes effectively local and the entanglement growth is thus eventually suppressed.

In contrast with that and analogously as for the case without the Hamiltonian, if the range $r$ of the jump operator is comparable with the total size $L$, the measurements can stabilize a volume-law scaling of the entanglement, as we can see in Fig. 2(c), referring to $r = L/2$. Here $\overline{S}_\ell$ quickly attains, in time, an asymptotic value that follows a volume-law scaling with $L$ [see the data in Fig. 2(f) for $\overline{S}_\ell/L$ versus $L^{-1}$, which display convergence to a finite value in the limit $L^{-1} \to 0$, and compare them with Fig. 1(f)].

It is finally worth commenting on the metastable plateaus displayed by the entanglement entropy, when considering ranges not extensively scaling with the system size. The curves reported in Fig. 3 are for $r = 1$ and correspond to those of Fig. 1(a) and Fig. 2(a), without and with the Hamiltonian, respectively. Here we changed the time axis according to $t \to \gamma t/L^2$ and noticed a fairly good convergence to an asymptotic scaling behavior with $L$. In fact, as highlighted in the two insets, the time $t^\star$ at which the entanglement entropy $S_\ell(t^\star)$ reaches a given threshold value scales as $t^* \sim L^2$.

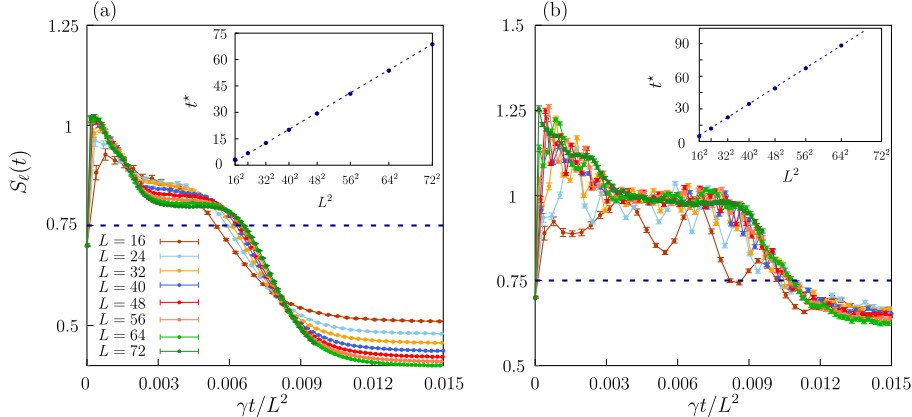

Figure 3: Same plots as in Fig. 1(a) and Fig. 2(a) for $r = 1$ (i.e., without and with the Hamiltonian $\hat{H}$, respectively), such that the time ($x$ axis) has been rescaled as $t \to \gamma t/L^2$. The two insets show the time $t^\star$ at which the entanglement entropy $S_\ell(t^\star)$ reaches the threshold value 0.75 (dashed line in the two main frames), as a function of $L^2$.

A less clear situation emerges at larger values of $r$. For example, when $r = 4$, with the numerical data at our disposal we could not find a clear signature of rescaling with $L$. Nonetheless, it emerges that the metastable plateau for $\hat{H} = 0$ has a lifetime that grows superlinearly with $L$.

## 5.4 Varying $\gamma$ and $r$

In the numerical analysis presented so far, the measurement rate has been always kept fixed. It is however quite natural that the parameter $\gamma$ should have an impact on the stabilization of the entanglement, when the measurement-induced dynamics competes with the (unitary) Hamiltonian dynamics. In fact, for fixed and finite values of $r$, we observe a change of behavior in the entanglement scaling with $L$, from a logarithmic growth to an area-law scaling, when varying $\gamma$ and in the presence of $\hat{H}$. An example of this is reported in Fig. 4(a) where we show, for $r = 1$, the asymptotic entanglement entropy $\overline{S}_\ell$ versus the system size $L$ for different measurement rates. In this case, a clear distinction between the two above mentioned regimes is observable for $\gamma_c \approx 0.15$. This behavior agrees with what has been already observed for a dynamics with onsite measurement operators [49,50]. To support this result in Fig. 4(b) we plot the rescaled entanglement entropy $\overline{S}/\log(L)$ vs $\gamma$ for different $L$. As expected, within the error bars, the curves collapse for $\gamma \lesssim 0.15$, indicating a logarithmic scaling. For larger $\gamma$, instead, the various curves drop, suggesting that the asymptotic entanglement entropy grows slower than $\log(L)$. On the other hand, for $r$ growing extensively with $L$, we expect that the volume-law scaling of the asymptotic entanglement entropy is robust.

Coming to the dependence of the entanglement dynamics on the range $r$ of the string measurement operators, differently from the onsite case, this should crucially depend on the system size $L$, since, as discussed above, the entanglement scaling is affected by the ratio $L/r$. Unfortunately, the numerical effort required to derive the results does not allow us to reach quantitative conclusions, because of the strong finite-size effects we have to deal with. However, our numerics suggests the existence of a critical value of $n = L/r$ ($n > 1$), separating a regime in which the asymptotic entanglement entropy does not change with the system size (i.e., $n \gg 1$), to a regime where it exhibits a linear growth with $L$ (i.e., $n \sim 1$).

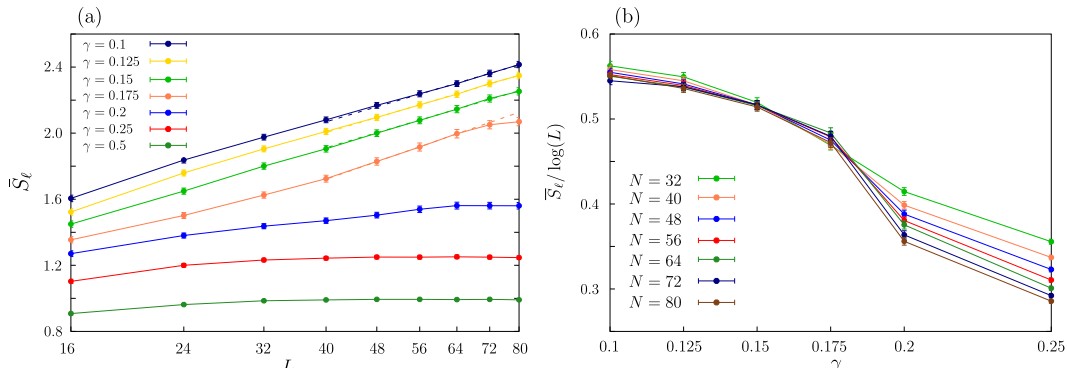

Figure 4: (a): Asymptotic entanglement entropy $\overline{S}_\ell$ versus the system size $L$, for $r = 1$ and different measurement rates (color scale). Here we fix $J = 1$ and quench the field from $h_i = 0.5$ to $h = 0.1$, as in Fig. 2. For $\gamma \lesssim 0.15$ we observe a logarithmic scaling, while for $\gamma \gtrsim 0.15$ an area-law behavior sets in. Note the logarithmic scale on the $x$ axis. (b): Rescaled asymptotic entanglement $\overline{S}/\log(L)$ vs $\gamma$. For $\gamma \lesssim 0.15$ the rescaled curves (are overlapped within the error bars), while they separate for larger measurement rates.

From our simulations, we cannot rule out the possibility that, in the thermodynamic limit, a volume-law scaling emerges for any non-diverging value of $n$. A qualitative argument supporting this possibility is the following. Assuming that two subsequent measurements with overlapping operators of range $r$ (which denotes the length of the corresponding string in spin-1/2 language) occur, these may strongly influence each other and thus generate large entanglement. Roughly speaking, the ratio between the probability $p_{\text{ov}}$ that the operators in subsequent measurements overlap and the probability $p_{\text{nov}}$ that they do not is $p_{\text{ov}}/p_{\text{nov}} \sim 2r/(L-2r)$. In the thermodynamic limit, this ratio remains nonvanishing only when $r = O(L)$ (i.e., $n \sim 1$). Thus it is reasonable to assume a volume-law scaling of the entanglement entropy whenever $L/r$ is not diverging.

We finally point out that, although the numerical analysis described above has been performed for a partition size fixed to $\ell = L/4$, we have done further checks to test the impact of this choice on our results. We found that the general scenario is qualitatively robust, at least for small and for large ranges $r$ of the string measurement operator, compared with $\ell$. Different behaviors, however, may appear when $r$ becomes of the order of the partition size $\ell$, suggesting that the important ratio to distinguish between the different entanglement scalings is $r/\ell$, rather than $r/L$.

## 6 Conclusions

We have unveiled some aspects of the entanglement-entropy dynamics of fermionic many-body Gaussian states, in the presence of a Gaussian-preserving evolution, with a special focus on the possible emergence of measurement-induced phase transitions in such systems. The first part of the paper presents a detailed technical discussion on how to maintain and treat Gaussianity, when exponentials of quadratic fermion operators are applied to a Gaussian state. Our purpose is to provide the reader with a comprehensive guide to implement fermionic Gaussian evolutions and study the dynamics (not necessarily unitary) of such systems. Among all the possible treatable evolutions, we focused on the quantum-jump unraveling of a Lindblad dynamics induced by a particular class of quadratic fermion operators, having a simple definition in the fermionic language, but a non-trivial string structure in terms of the Pauli matrices. We called them string operators.

In the second part of the paper, after having derived the equation of motions for a quantum-jump dynamics induced by the string operators, we have solved it for the quantum Ising chain, focusing on the time behavior of the entanglement entropy. Our main result is that the scaling of the entanglement entropy with the system size $L$ strongly depends on the range $r$ of the string operators. Because of numerical limitations we cannot investigate system sizes large enough to claim that this is an entanglement transition rather than a simple crossover. For short strings ($r$ finite), we see that the asymptotic entanglement entropy obeys a non-extensive scaling behavior. In this case, the measurement operator is effectively local in space and thus it asymptotically destroys quantum correlations that may be generated by the unitary dynamics. For string ranges comparable with the system size ($r \sim L$), the measurement process may become highly entangling and is able to stabilize a volume-law scaling in the asymptotic entanglement entropy.

Remarkably, the picture above is valid both for a measurement-only dynamics and for a dynamics where also the effect of a short-range Hamiltonian is considered. So, the measurement dynamics of the string operators is strong enough to induce different entanglement scalings by itself.

It would be tempting to address dynamical behaviors induced by more than a single type of string measurement operators (e.g., with different $r$), in such a way to explore the possibility of having frustration of the measurements [42, 74]. A further natural extension of our work could be to understand how the MIPT scenario would change in the presence of long-range measurement operators (e.g., by considering some power-law decay of dissipators with the distance). Finally, novel dynamical mechanisms may emerge in systems beyond the free-fermion paradigm, due to the role of interactions in the entanglement dynamics under variable-range monitoring.

## Acknowledgments

We acknowledge fruitful discussions with V. Alba, M. Collura, J. De Nardis, R. Fazio, M. Schirò, and X. Turkeshi.

**Funding information** This work has been partly funded by the Italian MIUR through PRIN Project No. 2017E44HRF and by PNRR MUR Project PE0000023-NQSTI.

## A   Rotation to the diagonal basis

Here we discuss a different derivation of the $s$-evolved state $|\psi(s)\rangle = \hat{M}(s)|\psi\rangle$, where $|\psi\rangle$ is a generic Gaussian state [Eq. (2)], and the $\hat{M}(s) = e^{\xi s \hat{A}}$ is defined in Eqs. (12).

The operator $\hat{A}$ [Eq. (12b)] is quadratic in the $\hat{c}$ fermions and can be written compactly as $\hat{A} = \hat{\Psi}^{\dagger} \mathbb{A} \hat{\Psi}$, where $\hat{\Psi} = (\hat{c}_1, \cdots, \hat{c}_L, \hat{c}_1^{\dagger}, \cdots, \hat{c}_L^{\dagger})^T$ is the Nambu spinor associated to the $\hat{c}$ fermions and $\mathbb{A}$ the Bogoliubov-de Gennes matrix associated to $\hat{A}$, as defined in Eq. (23). This matrix can be diagonalized by a Bogoliubov transformation $\mathbb{W}$, such that

$$\mathbb{W}^{\dagger} \mathbb{A} \mathbb{W} = \text{diag}[\epsilon_{\nu}, -\epsilon_{\nu}]_{\nu=1,\cdots,L}, \quad \text{and} \quad \hat{A} = \sum_{\nu} \epsilon_{\nu} \left( \hat{\eta}_{\nu}^{\dagger} \hat{\eta}_{\nu} - \hat{\eta}_{\nu} \hat{\eta}_{\nu}^{\dagger} \right). \tag{A.1}$$

The unitary matrix $\mathbb{W}$ defines a new set of $\hat{\eta}$ fermions, whose associated Nambu spinor is $\hat{\Theta} \equiv (\hat{\eta}_1, \cdots, \hat{\eta}_L, \hat{\eta}_1^{\dagger}, \cdots, \hat{\eta}_L^{\dagger})^T = \mathbb{W}^{\dagger} \hat{\Psi}$.

Let us now evaluate $|\psi(s)\rangle$. We remind that the initial state $|\psi\rangle$ is is the vacuum of the $\hat{\gamma}$ fermions, defined through the Bogoliubov transformation in Eq. (6): $\hat{\Phi} = \mathbb{U}^{\dagger} \hat{\Psi}$, where

$\hat{\Phi} = (\hat{\gamma}_1, \cdots, \hat{\gamma}_L, \hat{\gamma}_1^\dagger, \cdots, \hat{\gamma}_L^\dagger)^T$. These fermions can be transformed into those diagonalizing $\mathbb{A}$ by means of a rotation

$$\hat{\Phi} = \mathbb{U}^\dagger \mathbb{W} \hat{\Theta} = \mathbb{Q}^\dagger \hat{\Theta}, \quad \text{being} \quad \mathbb{Q} = \begin{pmatrix} u_\eta & v_\eta^* \\ v_\eta & u_\eta^* \end{pmatrix}. \tag{A.2}$$

Therefore we have the relation

$$\mathbb{U} = \mathbb{W} \mathbb{Q}, \tag{A.3}$$

with $\mathbb{U}$ given in Eq. (5). In the $\hat{\eta}$ representation, the state (2) takes the form

$$|\psi\rangle \propto \exp\left\{ \frac{1}{2} \sum_{\mu,\nu} Z_{\mu,\nu}^{(\eta)} \hat{\eta}_\mu^\dagger \hat{\eta}_\nu^\dagger \right\} |0\rangle_{\hat{\eta}}, \tag{A.4}$$

with $Z^{(\eta)} = -\left[ u_\eta^\dagger \right]^{-1} v_\eta^\dagger$.

Going on analogously as in Sec. 3.1, we can write the $s$-evolved state as

$$|\psi'\rangle \propto \exp\left\{ \frac{1}{2} \sum_{\mu,\nu} Z_{\mu,\nu}^{(\eta)} \hat{\eta}_\mu^\dagger(s) \hat{\eta}_\nu^\dagger(s) \right\} \hat{M}(s) |0\rangle_{\hat{\eta}}, \tag{A.5}$$

with $\hat{\eta}_\nu^\dagger(s) \equiv \hat{M}(s) \hat{\eta}_\nu^\dagger \hat{M}(-s)$. The operator $\hat{M}(s)$ is number-preserving, therefore we can exploit the methods of Section 4. We first notice that $\hat{M}(s) |0\rangle_\eta \propto |0\rangle_\eta$, in fact

$$e^{\xi s \sum_\nu \epsilon_\nu (\hat{\eta}_\nu^\dagger \hat{\eta}_\nu - \hat{\eta}_\nu \hat{\eta}_\nu^\dagger)} |0\rangle_{\hat{\eta}} = e^{-\xi s \sum_\nu \epsilon_\nu} |0\rangle_{\hat{\eta}}, \tag{A.6}$$

because $\hat{\eta}_\nu |0\rangle_{\hat{\eta}} = 0$, $\forall \nu \in \{1, \ldots, L\}$. Then we notice that, since $(\hat{\eta}_\nu^\dagger \hat{\eta}_\nu)^n = \hat{\eta}_\nu^\dagger \hat{\eta}_\nu$ (cf. Section 4), we have

$$\hat{M}(\pm s) = e^{\pm \xi s \hat{A}} = e^{\mp \xi s \sum_\nu \epsilon_\nu} \prod_\nu \left[ 1 + \sum_{n=1}^\infty \frac{(\pm 2\xi s \epsilon_\nu)^n}{n!} \hat{\eta}_\nu^\dagger \hat{\eta}_\nu \right]$$
$$= e^{\mp \xi s \sum_\nu \epsilon_\nu} \prod_\nu \left[ 1 + (e^{\pm 2\xi s \epsilon_\nu} - 1) \hat{\eta}_\nu^\dagger \hat{\eta}_\nu \right], \tag{A.7}$$

where we have applied the diagonal form of $\hat{A}$, in Eq. (A.1). Therefore, according to the fact that $\left[ \hat{\eta}_\mu^\dagger \hat{\eta}_\mu, \hat{\eta}_\nu^\dagger \right] = \delta_{\mu,\nu} \hat{\eta}_\mu^\dagger$, we can write

$$\hat{\eta}_\nu^\dagger(s) = \left[ 1 + (e^{2\xi s \epsilon_\nu} - 1) \hat{\eta}_\nu^\dagger \hat{\eta}_\nu \right] \hat{\eta}_\nu^\dagger \left[ 1 + (e^{-2\xi s \epsilon_\nu} - 1) \hat{\eta}_\nu^\dagger \hat{\eta}_\nu \right] = e^{2\xi s \epsilon_\nu} \hat{\eta}_\nu^\dagger. \tag{A.8}$$

Substituting into Eq. (A.5) we get

$$|\psi(s)\rangle \propto \exp\left\{ \frac{1}{2} \sum_{\mu,\nu} \tilde{Z}_{\mu,\nu}^{(\eta)} \hat{\eta}_\mu^\dagger \hat{\eta}_\nu^\dagger \right\} |0\rangle_{\hat{\eta}}, \quad \text{with} \quad \tilde{Z}_{\mu,\nu}^{(\eta)} \equiv Z_{\mu,\nu}^{(\eta)} e^{2\xi s (\epsilon_\mu + \epsilon_\nu)}. \tag{A.9}$$

We can also write $\tilde{Z}^{(\eta)} = -\left[ u_\eta^{\text{ev}\dagger}(s) \right]^{-1} v_\eta^{\text{ev}\dagger}(s)$, with

$$\begin{pmatrix} u_\eta^{\text{ev}}(s) \\ v_\eta^{\text{ev}}(s) \end{pmatrix} = \begin{pmatrix} e^{-2\xi s \epsilon} & 0 \\ 0 & e^{2\xi s \epsilon} \end{pmatrix} \begin{pmatrix} u_\eta \\ v_\eta \end{pmatrix}, \quad \text{being} \quad \epsilon = \text{diag}(\epsilon_\nu). \tag{A.10}$$

At this point, if $\xi = 1$, we apply the QR decomposition to $\begin{pmatrix} u_\eta^{\text{ev}}(s) \\ v_\eta^{\text{ev}}(s) \end{pmatrix}$, as explained in Sec. 3.2. In this way, the matrix $\tilde{Z}^{(\eta)}$ remains unchanged, but can be written as $\tilde{Z}^{(\eta)} = -\left[ u_\eta^{\ \dagger}(s) \right]^{-1} v_\eta^{\ \dagger}(s)$, with $u_\eta(s)$, $v_\eta(s)$ obeying the symplectic unitarity conditions Eq. (4) and written as

$$\begin{pmatrix} u_\eta^{\text{ev}}(s) \\ v_\eta^{\text{ev}}(s) \end{pmatrix} = \begin{pmatrix} u_\eta(s) \\ v_\eta(s) \end{pmatrix} r_1, \tag{A.11}$$

with $r_1$ some upper triangular matrix $L \times L$.

We have therefore reduced to the standard Gaussian form for the evolved state $|\psi(s)\rangle$ in the $\hat{\eta}$ representation, with the $Z$ matrix given by $\tilde{Z}^{(\eta)} = -\left[u_\eta{}^\dagger(s)\right]^{-1} v_\eta{}^\dagger(s)$, and the symplectic unitary matrix

$$\mathbb{Q}(s) = \begin{pmatrix} u_\eta(s) & v_\eta^*(s) \\ v_\eta(s) & u_\eta^*(s) \end{pmatrix}, \tag{A.12}$$

providing the set of fermionic operators $\hat{\gamma}_\nu(s)$ annihilating $|\psi(s)\rangle$ [we have $\hat{\Phi}(s) = \mathbb{Q}^\dagger(s)\hat{\Theta}$, with $\hat{\Phi}(s) \equiv (\hat{\gamma}_1(s), \cdots, \hat{\gamma}_L(s), \hat{\gamma}_1^\dagger(s), \cdots, \hat{\gamma}_L^\dagger(s))^T$]. At this point it is possible to come back to the $\hat{c}$ representation. We can use Eq. (A.3), and obtain the Bogoliubov matrix $\mathbb{U}(s)$ [the one such that $\hat{\Phi}(s) = \mathbb{U}^\dagger(s)\hat{\Psi}$] as $\mathbb{U}(s) = \mathbb{W}\,\mathbb{Q}(s)$. Applying $\mathbb{W}$ on the left of both sides of Eq. (A.10) and writing

$$\mathbb{U}(s) = \begin{pmatrix} u(s) & v^*(s) \\ v(s) & u^*(s) \end{pmatrix},$$

we obtain

$$\begin{pmatrix} u(s) \\ v(s) \end{pmatrix} r_1 = e^{-2\xi s \mathbb{A}} \begin{pmatrix} u \\ v \end{pmatrix} \tag{A.13}$$

[To get this result, we used the relation $\mathbb{U}(s) = \mathbb{W}\,\mathbb{Q}(s)$, as well as Eqs. (A.1), (A.3), and (A.11)]. The $Z$ matrix of the evolved state in this representation is thus $Z(s) = -\left[u^\dagger(s)\right]^{-1} v^\dagger(s)$. Notice that Eq. (A.13) is the same result of Eq. (24), when the QR decomposition discussed in Sec. 3.2 is applied.

As a last comment we point out that Eq. (A.13) refers to the first two blocks of the Bogoliubov matrix $\mathbb{U}$ only. This is consistent with the fact that, in the Nambu spinor notation, the degrees of freedom are doubled and there is some redundant information that can be neglected.

# B  Quantum-jump protocol

We start from the following master equation in the Lindblad form:

$$\dot{\rho}(t) = \mathcal{L}\big[\rho(t)\big] \equiv -i\big[\hat{H}, \rho(t)\big] - \gamma \sum_j \Big(\hat{m}_j \rho(t)\hat{m}_j^\dagger - \tfrac{1}{2}\{\hat{m}_j^\dagger \hat{m}_j, \rho(t)\}\Big), \tag{B.1}$$

where $\rho(t)$ is the density matrix describing the state of the system at time $t$ and $\hat{m}_j^{(\dagger)}$ are the Lindblad jump operators, while $\mathcal{L}$ denotes the Lindbladian superoperator acting on $\rho$. The solution of Eq. (B.1) is, in general, a mixed state that can be thought of as an average over many pure-state quantum trajectories of a suitable stochastic process. Given a Lindbladian, there are many stochastic processes (or unravelings) that produce the same averaged density matrix [10]. Among them, in this paper we focus on the so-called quantum-jump unraveling [83], whose implementation is described below.

We define the non-Hermitian Hamiltonian

$$\hat{H}_{\text{eff}} = \hat{H} - \frac{i\gamma}{2}\sum_j \hat{m}_j^\dagger \hat{m}_j, \tag{B.2}$$

then we rewrite Eq. (B.1)

$$\dot{\rho}(t) = -i\big[\hat{H}_{\text{eff}}, \rho(t)\big] - \gamma \sum_j \hat{m}_j \rho(t)\hat{m}_j^\dagger. \tag{B.3}$$

In this form, the Lindblad dynamics ruled by Eq. (B.1) can be interpreted as a deterministic non-Hermitian time evolution of a pure quantum state $|\psi_t\rangle$ generated by $\hat{H}_{\text{eff}}$, plus a stochastic part given by the possibility of applying $\hat{m}_j$ to the state $|\psi_t\rangle$ [cf. the second term in the right-hand-side of Eq. (B.3)]. For $\hat{m}_j$ Hermitian, this dynamics can be thought of as an occasional, yet abrupt, measurement process in which, at any time, the system has a chance to be POVM-measured [7,8], i.e., to undergo the action of one of the operators $\hat{m}_j$, taken from a distribution (see more details below).

In this framework, Eq. (B.1) can be solved by implementing the following stochastic process. The time evolution is discretized in time slices of step $dt$. Then, at each step:

1. With probability $\pi_j = \gamma\langle\hat{m}_j^\dagger\hat{m}_j\rangle dt$, the jump operator is applied:

$$|\psi_{t+dt}\rangle \mapsto \frac{\hat{m}_j|\psi_t\rangle}{||\hat{m}_j|\psi_t\rangle||}. \tag{B.4}$$

2. With probability $\pi = 1 - \sum_j \pi_j$ the non-Hermitian evolution occurs:

$$|\psi_{t+dt}\rangle \mapsto \frac{e^{-i\hat{H}_{\text{eff}}dt}|\psi_t\rangle}{||e^{-i\hat{H}_{\text{eff}}dt}|\psi_t\rangle||}. \tag{B.5}$$

Note that the Lindblad master equation in Eq. (B.1) is invariant under a shift of the jump operators $\hat{m}_j \mapsto \hat{1} + \hat{m}_j$. In practice, one can implement the protocol described above with the Lindblad operators $\hat{1} + \hat{m}_j$ (as we actually do in Sec. 5), instead of $\hat{m}_j$, and obtain the same averaged density matrix, although the single trajectories $|\psi_t\rangle$ may behave differently.

We remark that, since the von Neumann entropy of Eq. (54) quantifies the amount of genuine quantum correlations of a pure state, we first evaluate it over the single pure-state quantum trajectory and then perform the ensemble averaging. It is important to fix the order of the two operations of measuring and ensemble averaging, since they are commuting only for linear functions of the state $\rho$ (as for expectation values of observables), but not for quantities as the one in Eq. (54). For a more detailed discussion see, e.g., Refs. [44,49,50].

## C  Evolution of the state: A different derivation

In this appendix we provide a different derivation of the analytic expression for the evolved state $|\psi\rangle_{\hat{M}_{\mathcal{I},\mathcal{J}}}$, by evaluating the operator $\hat{\mathcal{Z}}(1) = \frac{1}{2}\sum_{p,q} Z_{p,q}\,\hat{c}_p^\dagger(1)\hat{c}_q^\dagger(1)$, in Eq. (16). To this purpose we notice that

$$\left[\hat{A}_{\mathcal{I},\mathcal{J}}, \hat{\mathcal{Z}}\right] = \frac{1}{2}\sum_{p,q}\sum_{i\in\mathcal{I}}\left(Z_{p,\mathcal{J}}\delta_{q,i} + Z_{q,\mathcal{J}}\delta_{p,i}\right)\hat{c}_p^\dagger\hat{c}_q^\dagger, \tag{C.1}$$

where we have defined $\hat{\tilde{\mathcal{Z}}} = \frac{1}{2}\sum_{p,q} Z_{p,q}\,\hat{c}_p^\dagger\hat{c}_q^\dagger$ and $Z_{p,\mathcal{J}} = \sum_{j\in\mathcal{J}} Z_{p,j}$. Consequently $\left[\hat{\tilde{\mathcal{Z}}}, [\hat{M}_{\mathcal{I},\mathcal{J}}, \hat{\tilde{\mathcal{Z}}}]\right] = 0$. This allows to write

$$\hat{M}_{\mathcal{I},\mathcal{J}}\,e^{\hat{\tilde{\mathcal{Z}}}} = e^{\hat{\tilde{\mathcal{Z}}}}\left(\hat{M}_{\mathcal{I},\mathcal{J}} + \left[\hat{M}_{\mathcal{I},\mathcal{J}}, \hat{\tilde{\mathcal{Z}}}\right]\right), \tag{C.2}$$

that, together with $\hat{M}_{\mathcal{I},\mathcal{J}}|0\rangle = |0\rangle$, leads to

$$\hat{M}_{\mathcal{I},\mathcal{J}}|\psi\rangle_{\hat{M}_{\mathcal{I},\mathcal{J}}} = \mathcal{N}\,e^{\hat{\tilde{\mathcal{Z}}}}\left(\hat{1} + \left[\hat{A}_{\mathcal{I},\mathcal{J}}, \hat{\tilde{\mathcal{Z}}}\right]\right)|0\rangle. \tag{C.3}$$

Since $\left[\hat{A}_{\mathcal{I},\mathcal{J}}, \hat{\tilde{\mathcal{Z}}}\right]^2 = 0$, we can write $\exp\left\{\left[\hat{A}_{\mathcal{I},\mathcal{J}}, \hat{\tilde{\mathcal{Z}}}\right]\right\} = \hat{1} + \left[\hat{A}_{\mathcal{I},\mathcal{J}}, \hat{\tilde{\mathcal{Z}}}\right]$ and, therefore,

$$\hat{M}_{\mathcal{I},\mathcal{J}}|\psi\rangle_{\hat{M}_{\mathcal{I},\mathcal{J}}} = \mathcal{N}\,e^{\hat{\tilde{\mathcal{Z}}}}e^{[\hat{A}_{\mathcal{I},\mathcal{J}}, \hat{\tilde{\mathcal{Z}}}]}|0\rangle \equiv \mathcal{N}\,e^{\hat{\tilde{\mathcal{Z}}}'}|0\rangle, \tag{C.4}$$

being

$$\hat{\bar{\mathcal{Z}}}' = \frac{1}{2}\sum_{i\in\mathcal{I}}\sum_{p,q}\Big[Z_{p,q} + \big(Z_{p,\mathcal{J}}\delta_{q,i} + Z_{q,\mathcal{J}}\delta_{p,i}\big)\Big]\hat{c}_p^\dagger\hat{c}_q^\dagger. \tag{C.5}$$

For the local case $\hat{M}_{\mathcal{I},\mathcal{J}}^{1}$ in Subsec 4.1, the jumped state $|\psi\rangle_{\hat{M}_{\mathcal{I},\mathcal{J}}^1}$ is described by the anti-symmetric matrix $\big[Z^1\big]'$, with

$$\big[Z^1\big]'_{p,q} = (1 + \delta_{q,i} + \delta_{p,i})Z_{p,q}. \tag{C.6}$$

On the other hand, for the non-local case $\hat{M}_{\mathcal{I},\mathcal{J}}^{\mathrm{nl}}$ in Subsec. 4.2, the antisymmetric matrix $\big[Z^{\mathrm{nl}}\big]'$ reads

$$\big[Z^{\mathrm{nl}}\big]'_{p,q} = Z_{p,q} + (\delta_{p,i} + \delta_{p,i+r})(Z_{i,q} + Z_{i+r,q}). \tag{C.7}$$

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
