# Peer review of "Entanglement dynamics with string measurement operators"

_SciPost Physics Core, doi:SciPost Phys. Core 6, 078 (2023)_

## Round 1 · Referee Report · Anonymous (Referee 1) · 2023-3-30

Strengths

  • Pedagogical discussion on some free fermionic techniques for non-unitary evolution

Weaknesses

  • Lack of novelty and impact (see report)
  • The physical aspects beyond the authors' model dynamics are only touched but not systematically investigated.

Report

Dear Editor,

This manuscript by G. Piccitto and collaborators reviews some known methods of non-unitary fermionic evolution. It further discusses some facets of quantum jump dynamics for long-range two-site fermionic measurements.

Brief Summary

In the first part (Sec. 2 to Sec. 4), the authors recall some general methodologies for non-unitary fermionic evolution. While pedagogical for the inexperienced reader or student, this discussion is nowadays well understood and typically present in any paper handling non-unitary free fermionic systems.

The second part of the manuscript (Sec. 5) briefly investigates the role of the two-body variable range $r$ fermions. The authors use entanglement entropy and discuss the effect of such measurements on (i) measurement-only dynamics and (ii) what happens when one also includes a Hamiltonian contribution.

In the former case, they demonstrate that the measurement only dynamics for $ r\sim~\xi$ ($\xi$ a fixed scale) saturate to an area-law entanglement scaling. Instead, the system can have extensive entanglement when $r\sim O(L)$. This fact is quite intuitive, as the projections are on arbitrary far Bell states (and motivates aspects of the measurement-based quantum computation). The authors further show that these aspects are somewhat resilient to the Hamiltonian addition. Finally, they conclude (Sec. 6) that a measurement-induced phase transition can occur.

Main novelties

The original contribution of the work is considering two-body variable range measurements on free fermionic models under a quantum jump evolution. The secondary novelty stems from explicating the details of the QR decomposition method to renormalize the wavefunction.

Recommendation

As I argue in the following, while some aspects of this paper deserve some form of publication when further analyzed, the paper lacks the novelty and impact requirement for publication in Scipost Physics (see https://scipost.org/SciPostPhys/about). Thus, I do not recommend the manuscript for publication in Scipost Physics.

In a nutshell, the "review" part of the manuscript is pretty understood. The technical details are reviewed in most papers on the field of monitored free fermions, and the details of the QR decomposition are a linear algebra exercise. Furthermore, the study of measurement-only dynamics has already been performed in various manuscripts (also in free fermionic systems, missing in the current bibliography). The main novelty is considering of long-range measurements. But I fail to see how this goes beyond "modeling" of monitored free fermions. This fact is further hindered by the lack of any analysis of the measurement-induced phase transition in terms of the range $r$ (i.e., the existence of a critical $r_c$), which is currently missing.

This missing analysis precludes me from suggesting the manuscript's present version in Scipost Physics Core (see https://scipost.org/SciPostPhysCore/about). However, if the authors can provide a study of how measurement-induced transitions appear varying $r$ as for the unitary counterparts, see Ref. [42,43], then I would be more than happy to suggest the paper for publication in Scipost Physics Core.

More detailed major criticisms and a list of minor criticisms are presented in the next section.

Criticisms

(Major 1): The non-unitary free fermionic techniques are fairly understood. Unitary and measurements-free fermionic systems are efficiently simulatable as detailed in the original contribution by Bravyi (e.g., Ref. [48]) and the extensive number of references in monitored free fermions (including missing references, e.g., [https://arxiv.org/abs/2210.05681, https://arxiv.org/abs/2012.04666,https://doi.org/10.1103/PhysRevResearch.4.033001,https://arxiv.org/abs/2205.07992,https://doi.org/10.1103/PhysRevB.104.184422,https://doi.org/10.1103/PhysRevResearch.4.L032026,https://arxiv.org/abs/2201.09895,https://doi.org/10.21468/SciPostPhys.14.3.031,https://arxiv.org/abs/2211.02534,https://arxiv.org/abs/2206.05384,https://quantum-journal.org/papers/q-2021-01-17-382,https://arxiv.org/abs/2302.09094,https://arxiv.org/abs/2111.08343], which I suggest including in the future version of the manuscript). The specific choice of simulations using the QR normalization has already been analyzed in U(1) symmetric systems (e.g., Ref. [33,35]) and in Majorana systems (e.g., Ref. [36,38,53]). (I also stress that an explicit derivation of the method has already appeared in one of the technical appendices of Ref. [53]. ) Summarising, I don't believe that reviewing these aspects gives an important addition to the field of monitored systems.

(Major 2): Measurement-only dynamics in free fermionic systems have already been analyzed (e.g., missing references [SciPost Phys. 14, 031 (2023), Phys. Rev. Lett. 126, 123604 (2021)] and Ref. [32,34]). The phenomenology for $r\sim O(L)$ follows the discussion in Ref. [31] for long-range measurements and, more generally, from basic aspects of measurement-based quantum computation. The only real aspect of interest is extending the volume law for finite r (as in Ref. [31]) to fermions. In this respect, any analysis is missing, and the data presented cannot justify the conjecture of a MIPT (as stated by the authors in the introduction) compared to a crossover behavior. While this would not be groundbreaking, such an analysis would make the manuscript publishable in Scipost Physics Core.

Let me raise a point to help further investigations of the authors in this respect: in Ref. [31], the frustration of the measurements seems a key aspect. Perhaps, if the MIPT is absent for only a single type of "string" measurements, the authors can consider two competing measures (e.g., see [ Phys. Rev. Lett. 126, 123604 (2021)]).

(Minor 1): The abstract is quite confusing. The "entanglement entropy of an Ising spin chain following a Lindblad dynamics with string measurement operators" may mislead the reader as it seems the authors are analyzing the entanglement entropy of a mixed state. The authors could rephrase that sentence. (Minor 2): Is the asymptotic entanglement obeying an area law for any choice of Hamiltonian? Not clear from the abstract and probably false (at least, possibly, for r=1). The authors should clarify this point. (Minor 3): In the introduction, the area law being constant seems specific to one-dimensional systems. The authors should add a small clarification. (Minor 4): I'm puzzled by the oscillation in fig 4 (b), surviving when measurements are present. Typically measurements cancel out these oscillations and lead to a fixed stationary state. Therefore, to me, it looks like the numerics may have some implementation problems. Can the authors provide a test for small systems (e.g., L=12, r=4, for the same parameters in Fig 2) comparing their fermionic simulations with exact diagonalization? (Minor 5): Missing ticks in the inset Fig 1b. Please add the ticks.

Requested changes

See report.

For publication in Scipost Physics Core: 1- the authors could systematically investigate the existence of a critical r, and draw a 2D phase diagram vs r and gamma at least for a choice of $h$. 2- resolve minor issues.

  • validity: ok
  • significance: low
  • originality: low
  • clarity: ok
  • formatting: good
  • grammar: good

Author:  Giulia Piccitto  on 2023-06-30  [id 3774]

(in reply to Report 1 on 2023-03-30)
Category:
remark

Please notice that in the resubmitted version you will find a clarification just before Eq. 25 (Section "QR-decomposition") that is not presented in the redline version we attached in the previous comment.

Author:  Giulia Piccitto  on 2023-06-27  [id 3766]

(in reply to Report 1 on 2023-03-30)
Category:
remark

Please find attached the reviewed version of the manuscript with all the corrections highlighted in red

Attachment:

lr_reply_09p2XEA.pdf

Author:  Giulia Piccitto  on 2023-06-27  [id 3763]

(in reply to Report 1 on 2023-03-30)
Category:
remark
answer to question
reply to objection
pointer to related literature

We thank the Referee for her/his careful reading of our manuscript and for all the useful
comments and suggestions. The explicit request for a numerical benchmark with the exact-
diagonalization method made us aware of a bug in the previous version of the code, which has
been now fixed. Below we provide an answer to all the raised points.

#1. The Referee writes:
The non-unitary free fermionic techniques are fairly understood. Unitary and measurements-free fermionic systems are efficiently simulatable as detailed in the original contribution by Bravyi (e.g., Ref. [48]) and the extensive number of references in monitored free fermions (including missing references [...], which I suggest including in the future version of the manuscript). The specific choice of simulations using the QR normalization has already been analyzed in $U(1)$ symmetric systems (e.g., Ref. [33,35]) and in Majorana systems (e.g., Ref. [36,38,53]). (I also stress that an explicit derivation of the method has already appeared in one of the technical appendices of Ref. [53].) Summarising, I don’t believe that reviewing these aspects gives an important addition to the field of monitored systems.

##Our answer:
The first part of the manuscript was intended as a pedagogical and detailed tool for people interested in the non-unitary dynamics of free-fermion systems. We are aware that, because of its review character, this part cannot be considered as an original contribution in the field of monitored systems. However we think that the way it is discussed can significantly help the general understanding of this nowadays hot topic (for example, even though it is a linear algebra exercise, we did not find any reference discussing the fundamental reason behind the QR-decomposition normalization techniques). To our understanding, our opinion is shared by the Second Referee, which counts our comprehensive review of non-unitary fermionic evolution as one of the strength points of the manuscript and found this part of the paper to be particularly well written and accessible to readers with limited experience in the field.
>Therefore we decided to keep this part as it is in the manuscript, after adding all the missing references quoted by the Referee (thanks to her/him for collecting and signaling them to us). See Refs. [25, 26, 34, 35, 36, 57, 58, 59, 60, 61, 67, 70, 72] in the revised version.

#2. The Referee writes:
Measurement-only dynamics in free fermionic systems have already been analyzed (e.g., missing references [SciPost Phys. 14, 031 (2023), Phys. Rev. Lett. 126, 123604 (2021)] and Ref. [32,34]). The phenomenology for $r \sim O(L)$ follows the discussion in Ref. [31] for long-range measurements and, more generally, from basic aspects of measurement-based quantum computation. The only real aspect of interest is extending the volume law for finite $r$ (as in Ref. [31]) to fermions. In this respect, any analysis is missing, and the data presented cannot justify the conjecture of a MIPT (as stated by the authors in the introduction) compared to a crossover behavior. While this would not be groundbreaking, such an analysis would make the manuscript publishable in Scipost Physics Core.

##Our answer:
We agree with the fact that the extensive scaling with $L$ of the entanglement entropy is somewhat justified by the presence of long-range monitoring. However, we think that this is a non trivial phenomenon concerning a timely and highly relevant research area, which deserves to be properly discussed in a journal as SciPost Physics. The relevance of the novel aspects covered by the manuscript, in the field of measurement-induced fermionic dynamics, is also recognized by the Second Referee. Having said this, we definitely agree with the Referee on the fact that an analysis of the entanglement dynamics as a function of $r$ would be highly desirable and is currently missing. We have performed several additional calculations and came to the conclusion that, unfortunately, the huge numerical effort required to go to sizes $L\gtrsim 100$ does not allow us to perform a systematic analysis for finite values of $r$. In particular, for large system sizes we need to go to very long times, since the convergence time typically scales with some power of $L$ larger than $1$ (we comment on this in the revised version), keeping small integration steps (that, we remind, for convergence reasons should scale as $\delta t \propto 1/\gamma L$). Therefore, we are not able to wash out large finite-size effects, which prevent us from locating sharp discontinuities with $r$ in the behavior of the entanglement entropy, and thus from drawing a proper phase diagram as a function of the range of the measurement operators. Given the importance of this point, we decided to extend the discussion in the main text (see, in particular, Sec. 5.4). We also rephrased the statement in the Introduction, to be more compatible with the presented data, avoiding any explicit reference to the presence of a measurement-induced phase transition with $r$. We added the suggested references as Refs. [58, 73].

#3. The Referee writes:
Let me raise a point to help further investigations of the authors in this respect: in Ref. [31], the frustration of the measurements seems a key aspect. Perhaps, if the MIPT is absent for only a single type of ”string” measurements, the authors can consider two competing measures (e.g., see [Phys. Rev. Lett. 126, 123604 (2021)]).

##Our answer:
We thank the Referee for this intriguing suggestion. We agree that frustration could be a key ingredient to investigate in our setup. One could consider a dynamics with two different kinds measurement operators, the first one connecting sites at a finite range, $A^{(1)}_j (r)$, and the second one with a range that scales with the system size, $A^{(2)}_j (L/n)$. Based on the numerics we performed for the present work, it is reasonable to believe that the resulting dynamics would be rather complex, due to the presence of many competing processes, and that it would require a large numerical effort. For this reason, we think that focusing on the frustration induced by the competition of different measurements goes beyond the purpose of this paper and thus prefer to leave it to further investigation somewhere else. We added a comment in the concluding section of the manuscript.

#4. The Referee writes (Minor):
The abstract is quite confusing. The “entanglement entropy of an Ising spin chain following a Lindblad dynamics with string measurement operators” may mislead the reader as it seems the authors are analyzing the entanglement entropy of a mixed state. The authors could rephrase that sentence.

##Our answer:
We clarified the statement in the abstract of the revised version.

#5. The Referee writes (Minor):
Is the asymptotic entanglement obeying an area law for any choice of Hamiltonian? Not clear from the abstract and probably false (at least, possibly, for $r = 1$). The authors should clarify this point.

##Our answer:
The Referee’s intuition is correct: for finite (and even for small) values of $r$, the asymptotic entanglement entropy may display logarithmic violations to an area-law behavior, depending on the values of the Hamiltonian parameters that govern the unitary dynamics and on the dissipation strength. The statement has been clarified in the revised version. See also the new Fig. 4, displaying a transition from area-law to logarithmic-law scaling with decreasing $\gamma$, for fixed Hamiltonian parameters and for $r = 1$.

#6. The Referee writes (Minor):
In the introduction, the area law being constant seems specific to one-dimensional systems. The authors should add a small clarification.

##Our answer:
We added a clarification to this point in the revised version.

#The Referee writes (Minor):
I’m puzzled by the oscillation in Fig 4 (b), surviving when measurements are present. Typically measurements cancel out these oscillations and lead to a fixed stationary state. Therefore, to me, it looks like the numerics may have some implementation problems. Can the authors provide a test for small systems (e.g., $L = 12$, $r = 4$, for the same parameters in Fig. 2) comparing their fermionic simulations with exact diagonalization?

##Our answer:
We thank the Referee for this observation, which allowed us to find a bug in the code that was responsible of quantitative (not qualitative) discrepancies of the previous results, with respect to the correct ones. In particular, we unwantedly exchanged two minus signs in the dissipative part of the evolution, that, from a physical point of view, is equivalent to exchange the “particle” and “antiparticle” contributions. Even though the qualitative behavior of the entanglement entropy is almost unaffected by this mistake, we observe some quantitative differences, especially on the transient behavior and the convergence times. The bug has been now fixed: In the figure in attachment we show a comparison between our corrected numerics (blue data set) and the exact-diagonalization results (green data set), for $L = 8$, $\gamma = 0.5$, $h = 0.1$, and $r = 1, 2, 3, 4$. The two data sets clearly coincide at any time, after making sure that the initial conditions are the same and that the same seed for the random numbers generator is used to produce the (non-deterministic) measurement-induced dynamics.

#The Referee writes (Minor):
Missing ticks in the inset Fig 1b. Please add the ticks.

##Our answer:
The figures of the paper have changed and ticks have been added.

Attachment:

match_exact.pdf

---

## Round 1 · Referee Report · Anonymous (Referee 2) · 2023-5-4

Strengths

  1. Active and highly relevant area of research.

  2. Good combination of analytic and numerical approaches.

  3. The paper provides a comprehensive review of non-unitary fermionic evolution.

  4. The paper investigates the existence of a volume law for finite r in a free fermion system, which is a relevant result.

Weaknesses

  1. Lack of a quantitative analysis of the MIPT in terms of gamma and r

Report

...Brief Summary...

The authors of the paper conducted a study on the impact of non-local string measurement operators on the dynamics of entanglement entropy in a quadratic Hamiltonian system. Their specific focus was on investigating the quantum-jump dynamics for the Ising model, using strings of Pauli matrices as Lindblad operators.

The key finding of the study was that, in contrast to the effects of local jump operators, the measurement process can stabilize a volume-law for the entanglement entropy when the string range r is comparable to the system size L. This implies that non-local measurements have a significant impact on the dynamics of entanglement entropy, particularly in the presence of quantum-jump dynamics.

... Structure of the Paper...

The authors of the paper began by providing a comprehensive review of the theory of quadratic fermionic Hamiltonians, and subsequently presented a detailed analysis of the dynamics of a Gaussian state under the influence of Gaussian-preserving exponentials of quadratic operators in Sections 2 to 4. I found this part of the paper to be particularly well written and accessible to readers with limited experience in the field.

In Section 5, the authors presented numerical results for the Ising chain, utilizing non-local string measurement operators for both measurement-only dynamics (neglecting the Hamiltonian contribution in Eq.(51)) and the full Lindblad Eq. (51). The findings of this section shed light on the impact of non-local measurement operators on the entanglement dynamics of the Ising model, and are likely to be of interest to researchers in the field of measurement-induced phase transitions.

... Major changes...

1) The paper lacks a quantitative analysis of the measurement-induced phase transition in terms of gamma. I would recommend that the authors conduct a study of the behavior of the entanglement entropy, while fixing r and varying gamma. Such an analysis would significantly enhance the validity of the paper.

2) A critical analysis of the measurement-induced phase transition in terms of r is currently lacking. It would be beneficial if the authors could study how the measurement-induced transitions appear, as the range r varies. Furthermore, the paper suggests the extension of the volume law for finite r. Unfortunately, no analysis has been provided to extrapolate the critical r. Such an analysis would significantly improve the quality of the manuscript and make it publishable for Scipost.

3) Furthermore, I would suggest that the authors include an analysis of how the phases of the model would change in the mutual presence of unitary dynamics and long-range measurement operators.

... Minor changes...

1) Missing ticks in the inset Fig 1b.

2) Fig 2(c) and Fig. 2(f) are not discussed in the text.

3) Specify the value of gamma used in the analysis presented in the manuscript.

4) Missing references 1) Phys. Rev. B 104, 184422 2)Phys. Rev. B 106, 024304 3) Nature Physics 17 (3), 342-347 4) Physical review letters 127 (23), 235701 5) arXiv:2210.05681 6) Phys. Rev. B 106 144313 (2022) 7) SciPost Phys. 14 031 (2023) 8) Physical Review Research 4 (2), 023179 9) Physical Review Research 2 (4), 043420 10) arXiv:2302.14551

5) Is it possible to investigate the scaling of S_l for different subsystem sizes?

Requested changes

  1. Extrapolation of critical r and gamma
  2. Phase diagram in function of r and gamma

  • validity: ok
  • significance: ok
  • originality: ok
  • clarity: good
  • formatting: good
  • grammar: good

Author:  Giulia Piccitto  on 2023-06-27  [id 3765]

(in reply to Report 2 on 2023-05-04)
Category:
remark

Please find attached the reviewed version of the manuscript with all the corrections highlighted in red

Attachment:

lr_reply.pdf

Author:  Giulia Piccitto  on 2023-06-27  [id 3764]

(in reply to Report 2 on 2023-05-04)
Category:
answer to question
reply to objection
correction
pointer to related literature

We thank the Referee for her/his careful reading of our manuscript and for all the useful comments and suggestions. Below we provide an answer to all her/his questions and comments.

#1. The Referee writes:
The paper lacks a quantitative analysis of the measurement-induced phase transition in terms of gamma. I would recommend that the authors conduct a study of the behavior of the entanglement entropy, while fixing r and varying gamma. Such an analysis would significantly enhance the validity of the paper.

##Our answer:
We have performed additional simulations to analyze what happens for fixed r and varying $\gamma$. We found that, at least for short measurement-operator ranges as $r = 1$, it is possible to identify a transition from a logarithmic scaling regime to an area-law one. This result is reasonable, especially if compared with the existence literature on the onsite measurement dynamics. We expect to find something similar for any fixed $r$, provided large enough system sizes are considered. We added this analysis and further discussions in the reviewed version of the manuscript (see Sec. 5.4).

#2. The Referee writes:
A critical analysis of the measurement-induced phase transition in terms of $r$ is currently lacking. It would be beneficial if the authors could study how the measurement-induced transitions appear, as the range $r$ varies. Furthermore, the paper suggests the extension of the volume law for finite $r$. Unfortunately, no analysis has been provided to extrapolate the critical $r$. Such an analysis would significantly improve the quality of the manuscript and make it publishable for Scipost.

##Our answer:
In fact, as also evidenced in Report 1, a thorough analysis of the putative measurement-induced transition in terms of $r$ is missing. Unluckily, as also explained in the reply above to the First Referee, after performing additional calculations we came to the conclusion that the huge numerical effort required to go to sizes $L \gtrsim 100$ does not allow us to perform a systematic analysis for finite values of $r$. In particular, for large system sizes we need to go to very long times, since the convergence time typically scales with some power of $L$ larger than 1 (we comment on this in the revised version), keeping small integration steps (that, we remind, for convergence reasons should scale as $\delta t \propto 1/\gamma L$). Therefore, we are not able to wash out large finite-size effects, which prevent us from locating sharp discontinuities with $r$ in the behavior of the entanglement entropy, and thus from drawing a proper phase diagram as a function of the range of the measurement operators. We have however added a critical discussion in the main text to explain this (see Sec. 5.4).

#3. The Referee writes:
Furthermore, I would suggest that the authors include an analysis of how the phases of the model would change in the mutual presence of unitary dynamics and long-range measurement operators.

##Our answer:
This is an interesting point. To answer this question, we first need to agree on the definition of the long-range measurement operators. If the question is about power-law decaying measurement operators as
\[
\hat m_j \sim \sum_{i\ne j} \frac{\hat c_i^\dagger \hat c_j + \hat c_j^\dagger \hat c_i}{|i-j|^\alpha},
\]
>we could have some technical difficulties, since the quantum-jump dynamics discussed in the main text cannot be straightforwardly extended to that case. However, we managed to write down a quantum-state-diffusion dynamics which preserves Gaussianity and accounting for such $\hat m_j$ operators. We are now starting to investigate a similar scenario: preliminary results suggest a very rich dynamics that we would rather discuss more thoroughly in a different paper. On the other hand, the easiest way to account for nonlocal measurements with multiple ranges $r \in \mathcal{R}$ and exploit the formalism introduced in the present manuscript is to write a generalized operator as
\[
\hat{\tilde{m}}_j = \big(\sum_{r\in \mathcal{R}} \hat c_{i+r}^\dagger \big)\big(\sum_{r’\in \mathcal{R}} \hat c_{i+r’}^\dagger \big)
\]
i.e., Eq. (37) in the manuscript, with $\mathcal{I} = \mathcal{J} \equiv \mathcal{R}$, which differs from the $\hat m_j$ written above. Technically, this dynamics can be simulated with our formalism, but it might be complicated to interpret it, due to the presence of different dynamical time scales related to the summation of operators on various sites. Novel competing effects, such as frustration, may set in (see also the reply to point 3 of the First Referee). For these reasons, we prefer to avoid adding such analysis to the manuscript.

#4. The Referee writes:
Missing ticks in the inset Fig 1b.

##Our answer:
The figure has been now replaced. The ticks should be ok now.

#5. The Referee writes:
Fig 2(c) and Fig. 2(f) are not discussed in the text.

##Our answer:
We added a discussion in the revised manuscript.

#6. The Referee writes:
Specify the value of gamma used in the analysis presented in the manuscript.

##Our answer:
We added it in the revised manuscript.

#7. The Referee writes:
Missing references [...]

##Our answer:
All the references mentioned by the Referee have been included in the revised version. See Refs. [23, 24, 27, 28, 29, 37, 57, 58, 60, 70] in the revised version.

#8. The Referee writes:
Is it possible to investigate the scaling of $S_\ell$ for different subsystem sizes?

##Our answer:
To investigate such scaling, we have done some calculations which suggest that the general scenario should not qualitatively change for small and for large values of $r$, although different behaviors may appear when r becomes of the order of the subsystem size $\ell$. A typical situation occurring when varying $\ell$ is displayed in the attached figure, which shows the time behavior of the entanglement entropy for $\ell = L/8, L/4, L/2$ (columns from left to right) and for $r = 1, 4, L/2$ (rows from top to bottom). Note that, while for all the three values of l we observe an area-law behavior for $r = 1$ and a volume-law behavior for r = L/2, the situation becomes more subtle for $r = 4$ (probably due to the limited range of sizes $L$ we are able to reach). In particular, when $r$ is of the same order of $\ell$ (central left panel) we observe a behavior similar to volume-law scaling; when r is much smaller than $\ell$ (central right panel) the curves for different $L$ get closer each other, thus suggesting a tendency to an asymptotic area-law scaling. We commented about this in the main text of the revised version.

Attachment:

ee_part_2.pdf

Author:  Giulia Piccitto  on 2023-06-30  [id 3775]

(in reply to Giulia Piccitto on 2023-06-27 [id 3764])
Category:
remark

Please notice that in the resubmitted version you will find a clarification just before Eq. 25 (Section "QR-decomposition") that is not presented in the redline version we attached in the previous comment.

---

## Round 2 · Author Response

'Following the questions and requests by the two Referees, we have performed additional simulations and produced an amended version of the manuscript.
For details see the reply to the Referees.
For details see the reply to the Referees.

---

## Round 2 · List of Changes

We changed Fig. 4 and the related discussion in the main text
We fixed minor issues
We updated some references with the published version
We fixed minor issues
We updated some references with the published version

---

## Editorial Decision

published